# TOWER: An Open Multilingual Large Language Model for Translation-Related Tasks

**Duarte M. Alves**[* 2,3]**, José Pombal**[* 1,2,3]**, Nuno M. Guerreiro**[* 1,2,3,4]
**Pedro H. Martins**[1]**, João Alves**[1]**, Amin Farajian**[1]**, Ben Peters**[2]**,**
**Ricardo Rei**[1]**, Patrick Fernandes**[2,6]**, Sweta Agrawal**[2]**,**
**Pierre Colombo**[4,5]**, José G.C. de Souza**[1] **& André F.T. Martins**[1,2,4]
duartemalves@tecnico.ulisboa.pt, {jose.pombal, nuno.guerreiro}@unbabel.com

## Abstract

While general-purpose large language models (LLMs) demonstrate proficiency on multiple tasks within the domain of translation, approaches based on open LLMs are competitive only when specializing on a single task. In this paper, we propose a recipe for tailoring LLMs to multiple tasks present in translation workflows. We perform continued pretraining on a multilingual mixture of monolingual and parallel data, creating TOWERBASE, followed by finetuning on instructions relevant for translation processes, creating TOWERINSTRUCT. Our model surpasses open alternatives on several relevant tasks and is competitive with general-purpose closed LLMs. We will release the TOWER models, our specialization dataset, an evaluation framework for LLMs focusing on the translation ecosystem, and a collection of model generations on our benchmark.

## 1 Introduction

Many important tasks within multilingual NLP, such as quality estimation, automatic post-edition, or grammatical error correction, involve analyzing, generating or operating with text in multiple languages, and are relevant to various translation workflows — we call these **translation-related tasks**. Recently, general-purpose large language models (LLMs) challenged the paradigm of *per-task* dedicated systems, achieving state-of-the-art performance on several recent WMT shared tasks (Kocmi et al., 2023; Freitag et al., 2023; Neves et al., 2023). Unfortunately, strong capabilities for *multiple* translation-related tasks have so far been exhibited by *closed* LLMs only (Hendy et al., 2023; Kocmi & Federmann, 2023; Fernandes et al., 2023; Raunak et al., 2023). Perhaps because most *open* LLMs are English-centric, approaches leveraging these models still lag behind, having thus far achieved competitive results only when specializing on a *single* task (Xu et al., 2024a; 2023; Iyer et al., 2023).

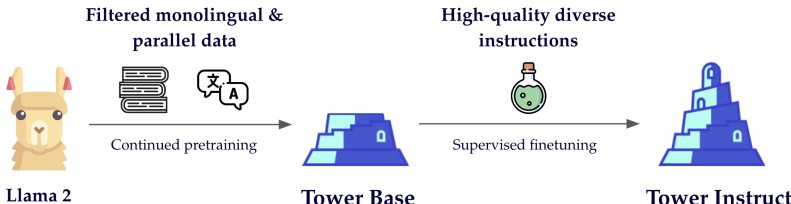

Figure 1: Illustration of our method for building TOWERBASE and TOWERINSTRUCT.

In this paper, we bridge this gap with a detailed recipe to develop an LLM for *multiple* translation-related tasks. Our approach, illustrated in Figure 1 and inspired by Xu et al.

[*]Equal Contribution: ordered alphabetically by first name. [1]Unbabel, [2]Instituto de Telecomunicações, [3]Instituto Superior Técnico & Universidade de Lisboa (Lisbon ELLIS Unit), [4]MICS, CentraleSupélec, Université Paris-Saclay, [5]Equall, [6]Carnegie Mellon University.

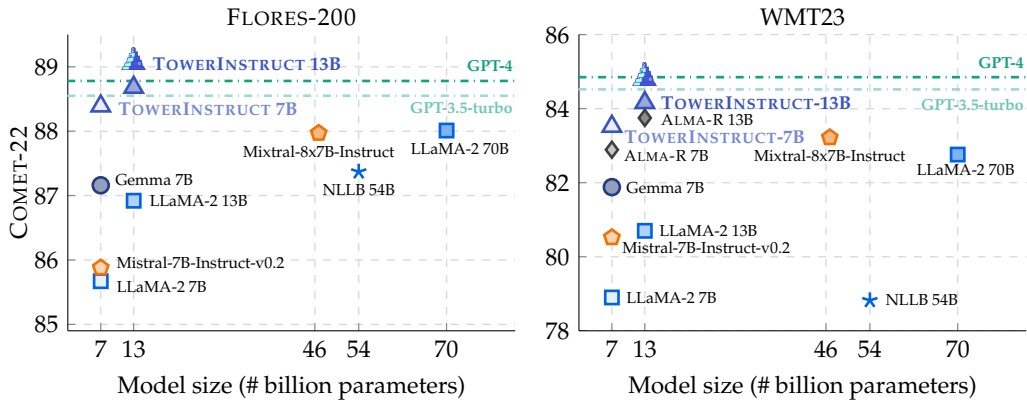

Figure 2: Translation quality for TOWERINSTRUCT and a collection of open and closed models. GPT models' scale is unknown so we represent them with a horizontal line. TOWERINSTRUCT outperforms larger open alternatives and is competitive with GPT models.

(2024a), relies on three steps. First, we extend LLaMA-2's (Touvron et al., 2023b) multilingual capabilities through continued pretraining, creating TOWERBASE (§2.1). Importantly, while Xu et al. (2024a) exclusively employ monolingual data, we include parallel data as an additional cross-lingual signal. Second, we curate a dataset to specialize LLMs for translation-related tasks, TOWERBLOCKS (§2.2). Third, we perform supervised finetuning to obtain an instruction-following model for the field of translation, TOWERINSTRUCT (§2.3).

We extensively evaluate all our models, comparing with open and closed alternatives on a wide range of tasks (§3). TOWERINSTRUCT consistently achieves higher translation quality than open alternatives and is competitive with the closed GPT-4 and GPT-3.5-turbo models — see Figure 2. Additionally, TOWERINSTRUCT outperforms open models in automatic post-edition, grammatical error correction, and named entity recognition. Careful ablations also outline the influence of each element in our recipe (§4). We highlight the importance of adding parallel data during continued pretraining for improved translation quality, and the effectiveness of including conversational and coding data on TOWERBLOCKS.

Accompanying this work, we release 1) the TOWER family, comprising our TOWERBASE and TOWERINSTRUCT models in the sizes of 7B and 13B; 2) our specialization dataset TOWERBLOCKS; 3) TOWEREVAL, the evaluation framework for LLMs for translation-related tasks that we used to perform all evaluations in this paper; and 4) a collection of model generations for our benchmark to ensure reproducibility and encourage future exploration.[1]

## 2 TOWER: An Open Multilingual LLM for Translation-Related Tasks

Our backbone language model is LLaMA-2 (Touvron et al., 2023b), which is very competitive on a wide range of tasks. Nevertheless, it was trained on relatively little non-English data, limiting its potential for multilingual tasks. We alleviate this effect by continuing LLaMA-2's pretraining on a highly multilingual corpus (§2.1). Afterwards, we finetune our continued pretrained model on a specialization dataset (§2.2), obtaining an instruction-following model centered around translation (§2.3).

### 2.1 TOWERBASE: Extending the multilingual capabilities of LLaMA-2

We extend LLaMA-2's training on a highly-multilingual dataset comprising 20 billion tokens — measured with the model's tokenizer — for 10 languages: English (en), German (de), French (fr), Dutch (nl), Italian (it), Spanish (es), Portuguese (pt), Korean (ko), Russian (ru),

---

[1]Links for the TOWER models; TOWERBLOCKS; TOWEREVAL; Zeno (Cabrera et al., 2023) project with model generations.

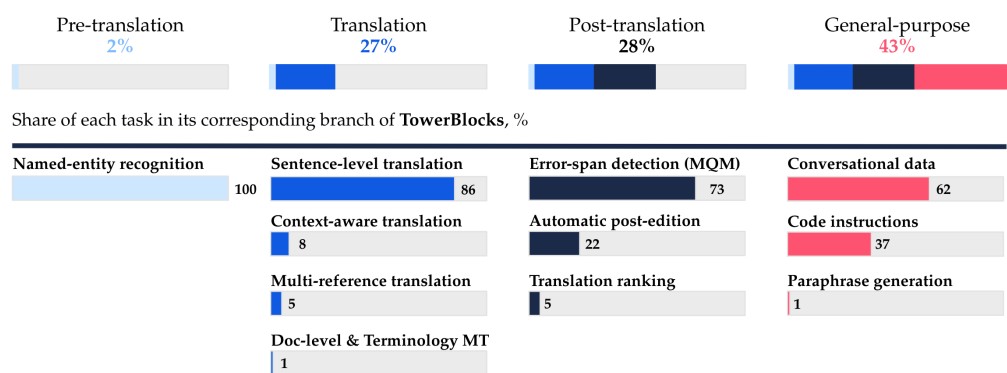

Figure 3: Tasks included in our supervised finetuning dataset TOWERBLOCKS.

and Chinese (zh). While Xu et al. (2024b) exclusively leverage monolingual data, we draw inspiration from work including parallel data during pretraining (Anil et al., 2023; Briakou et al., 2023), and *mix parallel sentences* (one third) along with monolingual data (two thirds). Our results show that this approach greatly benefits translation quality (§4).

**Monolingual data.** We sample uniformly across our languages from mC4 (Xue et al., 2021) and apply standard cleaning (Wenzek et al., 2019; Touvron et al., 2023a): deduplication, language identification, and perplexity filtering with KenLM (Heafield, 2011).

**Parallel Data.** We sample uniformly to-English (xx→en) and from-English (en→xx) language pairs from public sources, removing translations below quality thresholds for Bi-cleaner (Sánchez-Cartagena et al., 2018; Ramírez-Sánchez et al., 2020) as well as COMETKIWI-22 (Rei et al., 2022b), which is shown to improve translation quality (Peter et al., 2023). We include parallel data with the following template, calculating the loss on all tokens:

```
[source language]: [source]\n[target language]: [target]
```

**Model Training.** We train our models with a codebase based on Megatron-LLM (Cano et al., 2023) on 8 A100-80GB GPUs, using an effective batch size of 1.57 million tokens per gradient step and a cosine scheduler with initial and final learning rates of $3 \times 10^{-5}$ and $3 \times 10^{-6}$. We provide additional details on continued pretraining in Appendix D.1.

## 2.2 TOWERBLOCKS: A dataset to tailor LLMs for translation-related tasks

We build TOWERBLOCKS prioritizing data *diversity* and *quality*. Figure 3 illustrates all tasks in the dataset. They include tasks important to translation workflows, applied before or after translation, and datasets to improve multilingual understanding and instruction-following.

**Diversity.** We collect records from multiple datasets for each translation-related task — datasets are detailed in Appendix E. Akin to Wei et al. (2022), we reformulate all records as natural language instructions using multiple manually curated zero- and few-shot templates. Following insights from Longpre et al. (2023), we create zero-shot instructions with 75% of the records. For the remaining ones, we uniformly sample 1, 3, or 5 in-context examples from the respective dataset. We also include paraphrasing, dialogues from UltraChat (Ding et al., 2023), and code instructions from Glaive-Code-Assistant (Glaive AI, 2023).

**Quality.** Similarly to Xu et al. (2024a), we construct records from human-annotated datasets,[2] prioritizing validation and old test sets — we exclude datasets from 2023 onwards.

---

[2]For named entity recognition, we did not find a permissively licensed human-annotated dataset, so we use MultiCoNER (Malmasi et al., 2022; Fetahu et al., 2023). For general translation, we include a

For tasks with reference translations, such as translation and automatic post-edition, we score references with xCOMET-QE-ENSEMBLE (Guerreiro et al., 2023) and discard records with quality scores below 0.85. Additionally, we include only translation pairs in their original direction. Finally, we adopt the UltraChat (Ding et al., 2023) dialogues filtered by Tunstall et al. (2023) and further exclude records that contain translation requests, formatting issues (e.g. instructions starting with punctuation), or assistant refusal cases.

### 2.3 TOWERINSTRUCT: Specializing TOWERBASE for translation-related tasks

As a final step, we obtain TOWERINSTRUCT by finetuning TOWERBASE on TOWERBLOCKS.

**Dialog template.** We format each dialog as a single tokenizable string using the `chatml` template (Open AI, 2023). This template clearly separates between instructions and answers, and allows for multi-turn dialog. The template has three special identifiers (control tokens) to delimit messages: `<|im_start|>user` and `<|im_start|>assistant` precede the beginning of a turn, and `<|im_end|>` marks its end. We provide further details in Appendix F.1.

**Model training.** We finetune the model with the standard cross-entropy loss, enabling bfloat16 mixed precision and packing (Raffel et al., 2020). We only calculate the loss on target (answer) tokens. We train for 4 epochs using a low learning rate and a large batch size — we detail all hyperparameters in Appendix F.2.

## 3 Experiments

### 3.1 Experimental Setup

**Datasets and Tasks.** We analyze machine translation (MT) on FLORES-200 (NLLB Team et al., 2022), WMT23 (Kocmi et al., 2023), and TICO-19 (Anastasopoulos et al., 2020). Additionally, we examine three translation-related tasks. First, following Raunak et al. (2023), we evaluate automatic post-edition (APE) by measuring translation quality after post-editing NLLB-3.3B (NLLB Team et al., 2022) translations for WMT23. Second, we evaluate named entity recognition (NER), useful for entity anonymization, using the test split from Multi-CoNER 2023 (Fetahu et al., 2023).[3] Third, we evaluate grammatical error correction (GEC), which is *held out* from our training data and useful for correcting the source sentence before translation. We test GEC on CoNLL-2014 (Ng et al., 2014) (English), COWSL2H (Yamada et al., 2020) (Spanish), and mlconvgec2018 (Chollampatt & Ng, 2018) (German).

**Baselines.** We compare TOWER with the open LLaMA-2 70B (Touvron et al., 2023b) and Mixtral-8x7B-Instruct (Jiang et al., 2024), and the closed GPT-3.5-turbo and GPT-4.[4] For MT, we also consider dedicated translation models NLLB-54B (NLLB Team et al., 2022) and ALMA-R (Xu et al., 2024b). In Appendix G, we compare with the open Gemma 7B (Gemma Team, 2024), Mistral-7B-Instruct-v0.2 (Jiang et al., 2023) and Qwen1.5 72B (Bai et al., 2023).[5] Model generations use greedy decoding — we explore alternative decoding methods in Appendix A. We prompt TOWER and closed models in a 0-shot fashion and others with 5 examples randomly selected from the development set.

**Evaluation.** We evaluate translation quality with COMET-22 (Rei et al., 2022a) for both MT and APE. For MT, we also report xCOMET (Guerreiro et al., 2023), COMETKIWI-22 (Rei et al., 2022b), BLEURT (Sellam et al., 2020), and CHRF (Popović, 2015) in Appendix G.[6] For GEC, we

---

small amount of parallel data from OPUS to cover all language pairs. Nevertheless, we apply Bicleaner using a threshold of 0.85 followed by the quality filtering procedure described in this section.

[3]We uniformly sample 1000 of the more than 200k records due to the computational costs of evaluating all models on the whole test set.

[4]We use `gpt-3.5-turbo-0613` and `gpt-4-0613` available from the official OpenAI API.

[5]TOWERINSTRUCT outperforms all these open alternatives.

[6]Performance trends largely hold across metrics. Yet, there is a significant quality gap between ALMA-R and TOWER models in terms of CHRF — e.g., over 7 points in en→xx directions on WMT23.

| Models | FLORES-200 | | WMT 23 | | TICO 19 |
|---|---|---|---|---|---|
| | en→xx | xx→en | en→xx | xx→en | en→xx |
| **Closed** | | | | | |
| GPT-3.5-turbo | 88.95 2 | 88.14 3 | 85.56 2 | 83.48 2 | 87.36 2 |
| GPT-4 | **89.13** 1 | **88.42** 1 | **86.01** 1 | **83.69** 1 | **87.52** 1 |
| **Open** | | | | | |
| NLLB 54B | 86.79 4 | 87.95 3 | 78.60 7 | 79.06 6 | 87.05 2 |
| LLaMA-2 70B | 87.82 4 | 88.19 2 | 82.95 6 | 82.56 4 | 86.46 4 |
| Mixtral-8x7B-Instruct | 87.76 3 | 88.17 2 | 83.60 5 | 82.84 3 | 86.60 4 |
| ALMA-R 7B | — | — | 83.40 5 | 82.39 4 | — |
| ALMA-R 13B | — | — | 84.46 3 | 83.03 3 | — |
| TOWERINSTRUCT 7B | 88.51 3 | 88.27 2 | 84.28 3 | 82.77 4 | 87.01 3 |
| TOWERINSTRUCT 13B | 88.88 2 | **88.47** 1 | 85.14 2 | 83.18 2 | 87.32 2 |

Table 1: COMET-22 scores for translation test sets aggregated by language pairs. Models with statistically significant performance improvements are grouped in quality clusters. We highlight the best ranked models in bold and underline the best ranked open models.

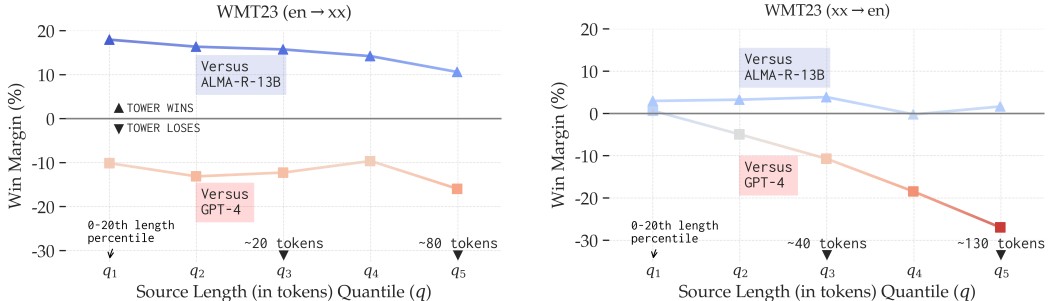

Figure 4: Win rates margin of TOWERINSTRUCT-13B by tokenized source length on WMT23 en→xx (left) and xx→en (right) directions. We compare against GPT-4 (□) and ALMA-R (△). We define a win if the sentence-level delta between two systems is above 1 COMET-22 point.

measure edit rate (ER) (Snover et al., 2006) and report ERRANT (Bryant et al., 2017; Felice et al., 2016) in Appendix H. For NER, we measure sequence F1 score. We report performance clusters based on statistically significant performance gaps at a 95% confidence threshold.[7] We create per-language groups for systems with similar performance, following Freitag et al. (2023), and obtain system-level rankings using a normalized Borda count (Colombo et al., 2022), which is defined as an average of the obtained clusters. Note that a first cluster will not exist if no model significantly outperforms all others on a majority of languages.

### 3.2 Translation

Table 1 reports aggregated results on translation test sets. Table 2 studies translation quality on all training languages, considering en→xx and xx→en translation directions.

**TOWERINSTRUCT 13B is the open system with highest translation quality.** TOWERINSTRUCT 13B consistently outperforms both larger open LLMs and dedicated systems. On FLORES-200, TOWERINSTRUCT 13B is often ranked first, and is close to GPT-4 performance

---

We posit that ALMA-R's alignment process on translations preferred by COMETKIWI-XXL (Rei et al., 2023) and xCOMET may inadvertently degrade performance on lexical metrics.

[7]For segment-level metrics, such as COMET-22, we perform significance testing at the segment level. For corpus-level metrics, such as ER and Sequence F1, we consider 100 bootstrapped samples of size 500, similar to Koehn (2004).

| Models | FLORES-200 (en→xx) | | | | | | | | |
|---|---|---|---|---|---|---|---|---|---|
| | de | es | fr | it | ko | nl | pt | ru | zh |
| **Closed** | | | | | | | | | |
| GPT-3.5-turbo | 88.78 2 | **87.08** 1 | **89.02** 1 | **89.06** 1 | 89.36 2 | **88.63** 1 | **90.46** 1 | 89.56 3 | 88.58 2 |
| GPT-4 | **88.98** 1 | **87.10** 1 | **88.93** 1 | **89.05** 1 | **90.06** 1 | **88.56** 1 | **90.43** 1 | **90.19** 1 | **88.87** 1 |
| **Open** | | | | | | | | | |
| NLLB 54B | 87.18 5 | 85.92 4 | 87.71 3 | 88.10 3 | 89.00 3 | 87.33 3 | 88.72 5 | 88.89 4 | 78.26 7 |
| LLaMA-2 70B | 87.31 5 | 86.41 3 | 87.82 3 | 88.22 3 | 88.07 4 | 87.47 3 | 89.11 4 | 88.65 5 | 87.32 5 |
| Mixtral-8x7B-Instruct | 87.99 3 | 86.80 2 | 88.53 2 | 88.77 2 | 85.63 5 | 87.57 3 | 89.45 3 | 89.09 4 | 85.99 6 |
| TOWERINSTRUCT 7B | 87.82 4 | 86.76 2 | 88.44 2 | 88.73 2 | 89.41 2 | 88.38 2 | 89.60 3 | 89.53 3 | 87.90 4 |
| TOWERINSTRUCT 13B | 88.16 3 | **87.06** 1 | **88.92** 1 | **89.21** 1 | **89.92** 1 | **88.63** 1 | 89.78 2 | 89.95 2 | 88.29 3 |

| Models | FLORES-200 (xx→en) | | | | | | | | |
|---|---|---|---|---|---|---|---|---|---|
| | de | es | fr | it | ko | nl | pt | ru | zh |
| **Closed** | | | | | | | | | |
| GPT-3.5-turbo | 89.60 2 | 87.26 3 | 89.46 3 | 88.03 3 | 87.83 3 | 87.71 2 | 89.78 3 | 86.69 4 | 86.92 2 |
| GPT-4 | **89.76** 1 | **87.57** 1 | **89.61** 1 | 88.21 2 | **88.58** 1 | **87.88** 1 | 89.94 2 | 86.94 2 | **87.29** 1 |
| **Open** | | | | | | | | | |
| NLLB 54B | 89.17 4 | 87.25 3 | 89.29 4 | 87.91 3 | 87.86 3 | 87.49 3 | 89.38 4 | 86.66 4 | 86.55 3 |
| LLaMA-2 70B | 89.44 3 | 87.49 2 | 89.55 2 | 88.18 2 | 87.91 3 | 87.52 3 | 89.84 2 | 86.87 2 | 86.91 2 |
| Mixtral-8x7B-Instruct | 89.57 2 | **87.65** 1 | 89.56 2 | **88.44** 1 | 87.37 4 | 87.54 3 | 89.73 3 | 86.81 3 | 86.88 2 |
| TOWERINSTRUCT 7B | 89.48 3 | 87.48 2 | 89.50 2 | **88.39** 1 | 88.16 2 | 87.66 2 | 89.92 2 | 86.90 2 | 86.96 2 |
| TOWERINSTRUCT 13B | 89.61 2 | **87.62** 1 | **89.67** 1 | **88.42** 1 | **88.48** 1 | **87.92** 1 | **90.07** 1 | **87.20** 1 | **87.27** 1 |

Table 2: Translation quality (via COMET-22) on FLORES-200 by language pair. Models with statistically significant performance are grouped in quality clusters. Best ranked models are highlighted in bold and best ranked open models are underlined.

on WMT23 and TICO-19. Upon inspecting both systems' outputs, we verified that the gap between them increases with longer sentences, as is shown in Figure 4. Notably, this trend vanishes when comparing TOWERINSTRUCT 13B to ALMA-R.[8] We posit this difference stems from a prevalence of shorter translations in the training data of both TOWERINSTRUCT 13B and ALMA-R. In future work, we would like to explore how to better leverage longer contexts, which can benefit instruction-following (Zhao et al., 2024).

**TOWERINSTRUCT 13B achieves high translation quality across all language directions.** In Table 2, TOWERINSTRUCT 13B is ranked first for the majority of en→xx directions, and is among the top performing models for all but one xx→en language pair. Notably, TOWERINSTRUCT stands out as the best overall model — outperforming GPT-4 — for both pt→en and ru→en directions. The improved performance on xx→en directions likely stems from LLaMA-2's English-centric pretraining. A longer, *more expensive* continued pretraining might further improve performance on en→xx directions. In fact, we show in Section 4 that the translation quality gains from LLaMA-2 are larger for en→xx language directions.

**TOWERINSTRUCT 7B achieves a trade-off between performance and scale.** The smaller TOWERINSTRUCT 7B, although behind TOWERINSTRUCT 13B, is competitive with other open systems and achieves GPT-3.5-turbo translation quality for some language pairs. Importantly, it outperforms the only system of the same size, ALMA-R 7B.

---

[8]A similar domain-level analysis did not find any domain dissimilar from the others.

| Models | APE (COMET-22)↑ en→xx | xx→en | GEC (ER)↓ Multilingual | NER (F1)↑ Multilingual |
|---|---|---|---|---|
| Baseline (no edits) | 76.80 | 79.99 | 16.66 | — |
| **Closed** | | | | |
| GPT-3.5-turbo | 81.47 4 | 78.68 5 | 15.06 2 | 50.22 4 |
| GPT-4 | **85.20** 1 | **84.30** 1 | 15.08 2 | 59.88 3 |
| **Open** | | | | |
| LLaMA-2 70B | 78.34 5 | 81.03 4 | 21.74 5 | 44.62 5 |
| Mixtral-8x7B-Instruct | 82.64 3 | 82.81 2 | 17.10 4 | 41.77 6 |
| TOWERINSTRUCT 7B | 82.69 2 | 81.56 4 | 15.13 3 | 71.68 2 |
| TOWERINSTRUCT 13B | 83.31 2 | 82.26 2 | 15.68 2 | **74.70** 1 |

Table 3: Results for translation-related tasks aggregated by language or language pair. Models with statistically significant performance improvements are grouped in quality clusters. We highlight the best ranked models in bold and underline the best ranked *open* models. Since GEC is a held out task, we evaluate all models with 5 in-context examples.

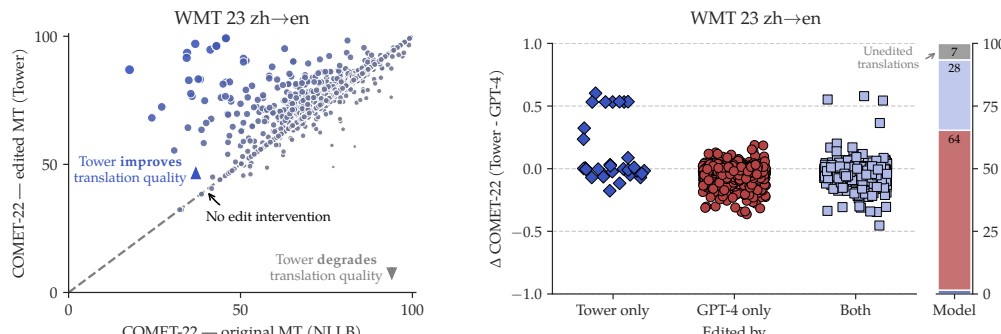

Figure 5: **Left:** Comparison between NLLB 3.3B original translation quality and TOWERINSTRUCT 13B post-edition quality on WMT23 zh→en. Marker size and hue represent the translation quality difference. **Right:** Post-edition quality difference on WMT23 zh→en when only TOWERINSTRUCT 13B edits (◇), only GPT-4 edits (○), or both models edit (□). The bar to the right represents the percentage of instances corresponding to each case.

## 3.3 Translation-Related Tasks

Table 3 reports aggregated results for all translation-related tasks.[9]

**TOWERINSTRUCT is an effective translation post-editor.** TOWERINSTRUCT outperforms open models and GPT-3.5-turbo on APE, consistently and significantly improving the quality of NLLB 3.3B translations. However, GPT-4 is still the top performer on this task. Yet it also has a higher edit rate, shown in Figure 5: while TOWERINSTRUCT edits 30% of the instances, GPT-4 edits almost 90%. This tendency suggests that GPT-4 is over-editing, which we further analyze in Appendix B. We posit that TOWERINSTRUCT edits less due to the prevalence of unedited segments in TOWERBLOCKS — roughly 38%.

**There is room for improvement on grammatical error correction.** On this task, no model significantly outperforms the others on a majority of languages. We hypothesize that TOWERINSTRUCT's average performance is caused by the absence of GEC data in TOWERBLOCKS.

---

[9]Appendix H details evaluated languages and provides further results for APE and GEC. Appendix C contains preliminary results for MT evaluation.

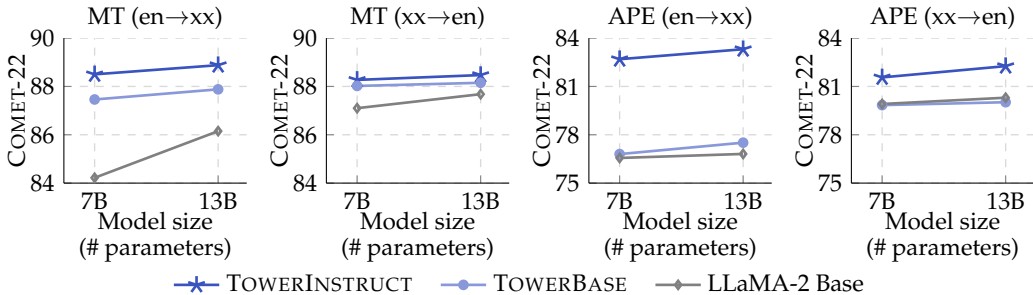

Figure 6: Recipe ablation across TOWER scales on MT (FLORES-200) and APE for en→xx and xx→en directions. We evaluate pretrained models (TOWER and LLaMA-2) in a 5-shot setting and TOWERINSTRUCT in a 0-shot fashion.

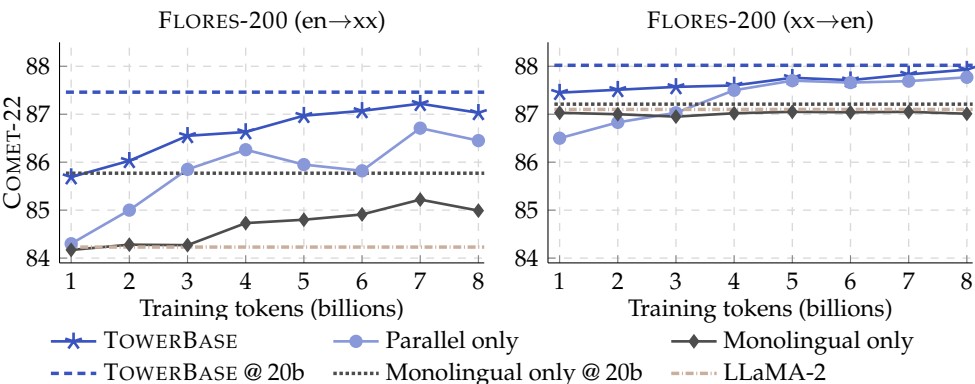

Figure 7: Translation quality on FLORES-200 for continue pretraining data recipes. The TOWERBASE recipe, outlined in Section 2.1, mixtures monolingual with parallel data. The "Parallel only" recipe only processed 8 billion tokens due to compute constraints.

**TOWERINSTRUCT can identify named entities.** TOWERINSTRUCT 13B shows promising performance on NER, surpassing GPT-4 by about 15 F1 points. Similar to APE, TOWERIN-STRUCT 7B reflects most of these improvements, highlighting its capabilities at a smaller scale. Other open models do not perform well on this task, even with 5 in-context examples. We hypothesize these results stem from NER being a token-level classification task, as opposed to a generative one. While the models learn the output format from the examples or task description, they struggle to grasp the classification function itself. Conversely, TOWERINSTRUCT learns the task from the NER data in TOWERBLOCKS.

## 4   Dissecting the training recipe

We performed multiple ablations to provide insights on the impact of the several design choices made in the development of the TOWER models.

**Continued pretraining and supervised finetuning yield independent performance gains.** The two leftmost plots of Figure 6 illustrate translation quality after continued pretraining and supervised finetuning. Both steps bring performance improvements at both model scales. Remarkably, TOWERBASE 7B and TOWERINSTRUCT 7B outperform LLaMA-2 13B, and TOWERINSTRUCT 7B outperforms TOWERBASE 13B. The two rightmost plots analyze APE. For this task, while supervised finetuning yields better performance, continued pre-training — and in particular parallel data — does not improve performance as observed for translation. In future work, we would like to explore additional training signals during continued pretraining to increase performance on translation-related tasks.

| Models | MT↑ | | APE↑ | | GEC↓ | NER↑ |
| | en→xx | xx→en | en→xx | xx→en | Multilingual | Multilingual |
|---|---|---|---|---|---|---|
| LLaMA-2 7B | 84.23 | 87.10 | 76.56 | 79.91 | 15.95 | 20.09 |
| TOWERBASE 7B | 87.46 | 88.02 | 76.79 | 79.83 | 15.41 | 20.51 |
| **Supervised Finetuning** | | | | | | |
| + MT | 88.45 | **88.28** | 79.19 | 79.36 | 54.76 | 0.00 |
| + Pre-MT + Post-MT | 87.92 | 87.96 | 81.95 | **81.73** | 17.44 | **74.92** |
| + General-Purpose | **88.51** | 88.27 | **82.69** | 81.56 | **15.13** | 71.68 |

Table 4: Ablation results for the components of TOWERBLOCKS. Results for pretrained models are obtained with 5 in-context examples while results for supervised models are obtained in a 0-shot setup. We consider FLORES-200 to evaluate translation quality.

**Parallel data during continued pretraining improves translation quality.** Figure 7 reports 5-shot translation quality on FLORES-200 for multiple continued pretraining data recipes. Mixing monolingual and parallel data achieves the highest quality, outperforming both monolingual-only and parallel-only data mixes. In general, improvements are more noticeable on en→xx directions, likely due to the English-centric nature of LLaMA-2's training. Nevertheless, while monolingual-only data improves over the base LLaMA-2 by 0.1 COMET-22 points on xx→en directions, our recipe gains nearly a full point.[10]

**Parallel data during continued pretraining is sample efficient, but quality continues to improve with more tokens.** At the 2 billion token mark, combining parallel sentences with monolingual data (i) yields more than 50% of the improvement over the base model, and (ii) surpasses the recipe leveraging solely monolingual data. Additionally, although training on more tokens has diminishing returns — 85% of the total performance gains appear by the 5 billion token mark — it continues to improve translation quality.

**Transfer/interference relations between tasks are complex.** Table 4 ablates the components of TOWERBLOCKS, comparing finetuning on translation data, translation-related tasks including pre- and post-translation, and the full dataset with general-purpose tasks. While adding translation-related tasks improves their performance, it decreases translation quality. Remarkably, introducing general-purpose instructions recovers translation quality. In future work, we would like to explore transfer/interference between tasks using scaling laws.

**The TOWER recipe generalizes to other model families.** When newer LLMs become available, opportunities for improving TOWER naturally arise. In Table 5 we compare Mistral 7B against LLaMA-2 as a backbone for TOWER. Notably, applying the TOWER recipe to Mistral outperforms starting from LLaMA-2 7B across the board and is competitive with using the larger LLaMA-2 13B, showcasing the generalizability of the recipe.[11]

## 5 Related Work

Adapting open models to *single* tasks within the field of machine translation is competitive with closed models or dedicated systems (Xu et al., 2024a; 2023; Iyer et al., 2023). Notably, Xu et al. (2024a) adapt LLaMA-2 for translation with continued pretraining on *monolingual* data and finetuning on high quality parallel data. Our work adopts a similar approach, but introduces *parallel* data during continued pretraining and leverages LLMs' instruction-following capabilities to build a system supporting *multiple* translation-related tasks.

---

[10]While 0.1 COMET-22 points translates to 54.9% human agreement, one COMET-22 point translates to 90.9% (Kocmi et al., 2024).

[11]We also release our TOWER models based on Mistral.

| Backbone Models | MT↑ | | APE↑ | | GEC↓ | NER↑ |
| --- | --- | --- | --- | --- | --- | --- |
| | en→xx | xx→en | en→xx | xx→en | Multilingual | Multilingual |
| LLaMA-2 7B | 88.51 | 88.27 | 82.69 | 81.54 | 15.13 | 71.68 |
| LLaMA-2 13B | 88.88 | **88.47** | **83.31** | **82.26** | 15.68 | **74.70** |
| Mistral 7B | **88.98** | 88.44 | 83.05 | 81.95 | **14.70** | 72.54 |

Table 5: Results for translation-related tasks aggregated by language or language pair among different backbone models trained on the TOWER recipe. We consider FLORES-200 to evaluate translation quality.

**Multilinguality in LLMs.** Previous work building more multilingual LLMs either trains a new model "from scratch" (Üstün et al., 2024; Faysse et al., 2024; Wei et al., 2023), or extends the pretraining of an existing model, possibly with vocabulary extension (Cui et al., 2023; Xu et al., 2024a; Pires et al., 2023). Building upon the effectiveness of pretraining on parallel data (Anil et al., 2023; Wei et al., 2023), we include *parallel* sentences during continued pretraining. We do not extend the vocabulary, as preliminary experiments yielded negative results.

**Specialization of LLMs.** Recent research on specializing LLMs on closely-related tasks leverages domain-specific data to train new models "from scratch" (Taylor et al., 2022; Wu et al., 2023), extend the pretraining of existing models (Lewkowycz et al., 2022; Chen et al., 2023), perform supervised finetuning (Yue et al., 2024), or a combination of the last two (Rozière et al., 2023; Liu et al., 2023). Our specialization approach is broadly inspired by instruction tuning (Wei et al., 2022; Sanh et al., 2022; Wang et al., 2023; Zhou et al., 2023), which finetunes language models on tasks formatted as natural language instructions.

## 6 Conclusion

We propose a new recipe for specializing LLMs on *multiple* translation-related tasks. First, we expand LLaMA-2's multilingual capabilities with continued pretraining on a highly multilingual corpus. Then, we finetune on a dataset of high-quality and diverse instructions for translation-related tasks. Our final model consistently outperforms *open* alternatives on multiple translation-related tasks, and is competitive with *closed* models such as GPT-4.

### Acknowledgments

We thank António Farinhas and Manuel Faysse for the fruitful discussion throughout the project. Part of this work was supported by the EU's Horizon Europe Research and Innovation Actions (UTTER, contract 101070631), by the project DECOLLAGE (ERC-2022-CoG 101088763), by the Portuguese Recovery and Resilience Plan through project C645008882-00000055 (Center for Responsible AI), and by Fundação para a Ciência e Tecnologia through contract UIDB/50008/2020. We also thank GENCI-IDRIS for the technical support and HPC resources — Jeanzay grants 101838, 103256, 103298 and Adastra grants C1615122, CAD14770, CAD15031 — used to partially support this work.

## 7 Ethics Statement

Our work builds upon LLMs to develop an instruction-following model centered around translation. The usage of LLMs encompasses several risks, which are thoroughly discussed in Brown et al. (2020) and Chowdhery et al. (2022). While we performed extensive evaluation on our models, we focused on performance metrics and did not include toxicity or safety analysis. Additionally, we leveraged automatic metrics which are trained on human annotations and can inherit issues from mislabeled data. Annotators can fail to consider better alternatives when presented with a given text or wrongfully classify text as high quality (Bansal et al., 2021).

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

# A   Analysis of alternative decoding strategies

| Models | FLORES-200 | | WMT 23 | | TICO 19 |
|---|---|---|---|---|---|
| | en→xx | xx→en | en→xx | xx→en | en→xx |
| GPT-3.5-turbo | 77.08 | 78.12 | 72.06 | **72.50** | 75.91 |
| GPT-4 | 77.26 | 78.51 | **72.54** | **72.91** | **76.16** |
| TOWERINSTRUCT 13B | | | | | |
| Greedy | 76.89 | 78.67 | 70.87 | 71.75 | 75.40 |
| Beam | 77.40 | **78.87** | 71.31 | 71.88 | 75.66 |
| MBR | **77.79** | **78.96** | 72.29 | 72.36 | 76.13 |

Table 6: Impact of beam search and minimum Bayes risk (MBR) decoding in translation quality for TOWERINSTRUCT 13B. In bold, we highlight systems in the first quality cluster. For TICO-19 there is no first cluster since no model significantly outperforms the others on a majority of the language pairs.

In this section, we analyse the performance of TOWERINSTRUCT 13B with beam-search (Reddy, 1977) using beam size of 5 and minimum Bayes risk (MBR) decoding (Eikema & Aziz, 2020; Fernandes et al., 2022; Freitag et al., 2022) with 20 hypotheses and COMET-22 as an utility function. We generate hypotheses using temperature and nucleus sampling (Holtzman et al., 2020), with $t = 0.9$ and $p = 0.6$. We avoid "optimizing" the evaluation metric (Fernandes et al., 2022) by measuring translation quality with BLEURT.

Table 6 reports translation quality across all test sets. Both decoding strategies consistently improve translation quality over greedy decoding, with MBR decoding achieving higher quality. Additionally, for both WMT23 and TICO-19, decoding strategies close the gap to GPT-4. Notably, on FLORES-200, TOWERINSTRUCT 13B appears isolated in the first cluster.

# B   Further analysis on TOWERINSTRUCT and GPT-4 editing tendencies

Figure 8 shows that differences between GPT-4 and TOWERINSTRUCT edit rates are not strongly correlated to differences in COMET-22 (0.34 Spearman $\rho$). This means that GPT-4 edits often do not correspond to gains in performance. This finding, allied with the discussion in Section 3.3 about GPT-4 editing considerably more than TOWERINSTRUCT, suggests that GPT-4 may be editing too much.

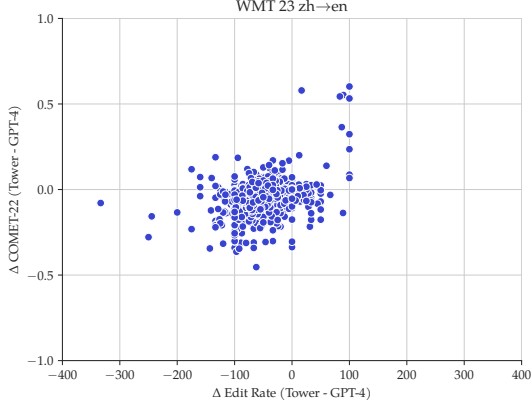

Figure 8: Difference between TOWERINSTRUCT 13B and GPT-4 edit rate (compared to the original NLLB translation) (x-axis), and difference between TOWERINSTRUCT 13B and GPT-4 post-edition COMET-22 (y-axis). The correlation between the two variables is 0.34 Spearman $\rho$. Similar patterns are observed for other language pairs.

|         | WMT23 | |
| Models  | en→de | zh→en |
| --- | --- | --- |
| COMET-22 | 0.3819 | 0.2731 |
| TOWER | **0.3994** | **0.3292** |

Table 7: Kendall τ correlation of reference-based MT evaluation metrics with human judgements on WMT23 data.

## C TOWERINSTRUCT as an evaluator of translation quality.

We included evaluation data in TOWERBLOCKS to induce positive transfer to machine translation. Nevertheless, recent work has shown that LLMs can also function as estimators translation quality (Kocmi & Federmann, 2023; Fernandes et al., 2023). Table 7 shows preliminary experiments where the evaluation data in TOWERBLOCKS is improved. Notably, this version of TOWER is competitive with COMET-22 on en→de and zh→en, suggesting that the TOWER recipe can be used to create models to evaluate translation quality. In the future, we intend to further explore how to improve evaluation capabilities of TOWERINSTRUCT.

## D Details of continued pretraining

This section contains some details on the continued pretraining process used to obtain TOWERBASE. Training time was 10 and 20 days for the 7B and 13B models, respectively.

### D.1 Filtering thresholds

In Table 8, we report the perplexity floors and ceilings used to filter the monolingual data in the continued pretraining corpus, as well as the Bicleaner and CometKiwi-22 thresholds used to filter the parallel data.

### D.2 Parallel data sources

Parallel data was obtained from the sources listed in Table 9.

|  | en | de | fr | nl | es | pt | ru | zh | ko |
| --- | --- | --- | --- | --- | --- | --- | --- | --- | --- |
| Min. perplexity ∗ | 50 | 50 | 50 | 50 | 50 | 50 | 50 | 50 | 50 |
| Max. perplexity ∗ | 516 | 611 | 322 | 649 | 275 | 257 | 334 | 2041 | 198 |
| Bicleaner † | - | 0.5 | 0.5 | 0.5 | 0.5 | 0.5 | 0.5 | 0.0 | 0.5 |
| COMETKIWI-22 † | - | 0.75 | 0.75 | 0.75 | 0.75 | 0.75 | 0.75 | 0.75 | 0.75 |

Table 8: Quality filtering thresholds applied on monolingual data (∗) and parallel data (†) by language. On the latter, the to-English language pair's threshold is the same as the corresponding from-English one.

| Dataset | Version |
|---------|---------|
| Europarl (Koehn, 2005) | v8 |
| ParaCrawl (Esplà et al., 2019) | v9 |
| MultiParaCrawl (Esplà et al., 2019) | v7.1 |
| CCMatrix (Schwenk et al., 2020) | v1 |
| CCAligned (El-Kishky et al., 2020) | v1 |
| MultiCCAligned (El-Kishky et al., 2020) | v1 |
| WikiTitles (Tiedemann, 2012) | v2014 |
| WikiMatrix (Schwenk et al., 2019) | v1 |
| News-Commentary (Tiedemann, 2012) | v16 |
| OPUS100 (Zhang et al., 2020) | v1 |
| TildeModel (Rozis & Skadiņš, 2017) | v2018 |
| Bible (Mayer & Cysouw, 2014) | v1 |
| Ubuntu (Tiedemann, 2012) | v14.10 |
| Tatoeba (Tiedemann, 2012) | v2 |
| GNOME (Tiedemann, 2012) | v1 |
| GlobalVoices (Tiedemann, 2012) | v2018q4 |
| KDE4 (Tiedemann, 2012) | v2 |
| KDE-Doc (Tiedemann, 2012) | v1 |
| PHP (Tiedemann, 2012) | v1 |
| Wikipedia (Wołk & Marasek, 2014) | v1.0 |
| Wikimedia (Tiedemann, 2012) | v20210402 |
| JRC (Tiedemann, 2012) | v3.0 |
| DGT (Tiedemann, 2012) | v2019 |
| EuroPat (Europat) | v3 |
| EUbookshop (Tiedemann, 2012) | v2 |
| EMEA (Tiedemann, 2012) | v3 |
| EUConst (Tiedemann, 2012) | v1 |
| tico-19 (Anastasopoulos et al., 2020) | v20201028 |
| ECB (Tiedemann, 2012) | v1 |
| Elitr-ECA (Williams & Haddow, 2021) | v1 |
| MultiUN (Eisele & Chen, 2010) | v1 |
| OpenOffice (Tiedemann, 2012) | v3 |
| Ada83 (Tiedemann, 2012) | v1 |
| infopankki (Tiedemann, 2012) | v1 |
| Scielo (Soares et al., 2018) | v1 |
| giga-fren (Tiedemann, 2012) | v2 |
| UNPC (Ziemski et al., 2016) | v1.0 |

Table 9: The various data sources used to create the parallel data with the number of available language pairs.

# E   Details of TOWERBLOCKS

This appendix details all datasets utilized in TOWERBLOCKS:

- **WMT14 to WMT21**[12] — Evaluation sets for the general machine translation shared task;
- **WMT22 with quality-shots** (Hendy et al., 2023) — Evaluation set from WMT23 with high quality in-context examples;
- **NTREX** (Federmann et al., 2022) — Professional translations of the WMT19 test set;
- **FLORES-200** (NLLB Team et al., 2022) — Development set of the FLORES-200 dataset for all languages included in training;
- **FRMT** (Riley et al., 2022) — Human translations of English Wikipedia sentences into regional variants;
- **OPUS** (Tiedemann, 2012) — Parallel corpora from which we sampled very high-quality samples for all language pairs;
- **QT21** (Specia et al., 2017) and **ApeQuest**[13] — Translation data with post-edits utilized for general translation and automatic post-editing;
- **MT-GenEval** (Currey et al., 2022) — Gender translation benchmark which we leveraged for general translation and context-aware translation;
- **WMT20 to WMT22 Metrics MQM**[14] — MT evaluation data annotated with multidimensional quality metrics (Lommel et al., 2014) that we used to perform error span detection;
- **WMT17 to WMT22 Metrics DAs**[15] — MT evaluation data annotated with direct assessements (DAs) (Graham et al., 2013) which we utilized for translation ranking.
- **WMT21 Terminology**[16] — Development set for the WMT21 terminology task;
- **Tatoeba** (Tiedemann, 2020) — Development set of the Tatoeba dataset which we used to generate translations in different languages for the same source — we identified this task as multi-reference translation;
- **MultiCoNER 2022 and 2023** (Malmasi et al., 2022; Fetahu et al., 2023) — Development sets of the named entity recognition MultiCoNER datasets. For MultiCoNER 2023, we adopted the coarse-grained entity categorization;
- **PAWS-X** (Yang et al., 2019) — Development set of the PAWS-X dataset which we used as paraphrase generation;
- **UltraChat** (Ding et al., 2023) — Filtered version of the UltraChat dataset used in Tunstall et al. (2023);
- **Glaive Code Assistant** (Glaive AI, 2023) — Coding questions and answers across a wide range of programming languages.

---

[12]https://www2.statmt.org/wmt23/translation-task.html

[13]https://apequest.wordpress.com/

[14]https://www.statmt.org/wmt22/results.html

[15]https://www.statmt.org/wmt22/results.html

[16]https://www.statmt.org/wmt21/terminology-task.html

# F  Details of TOWERINSTRUCT

This appendix further details the supervised finetuning procedure to train TOWERINSTRUCT.

## F.1  Chat Template

We finetuned TOWERINSTRUCT with the chatml template (Open AI, 2023). Table 10 provides an example of an interaction using the aforementioned template.

| | |
|---|---|
| **User** | `<\|im_start\|>user`
Translate the following text from Portuguese into English.
Portuguese: Ontem, a minha amiga foi ao supermercado mas estava fechado. Queria comprar legumes e fruta.
English: `<\|im_end\|>`
`<\|im_start\|>assistant` |
| **Model** | Yesteday, my friend went to the supermarket but it was closed. She wanted to buy vegetables and fruit.`<\|im_end\|>` |
| **User** | `<\|im_start\|>user`
Can you now translate it into Spanish? `<\|im_end\|>`
`<\|im_start\|>assistant` |
| **Model** | Ayer mi amiga fue al supermercado, pero estaba cerrado. Quería comprar verduras y fruta.`<\|im_end\|>` |

Table 10: Example of a dialogue with TOWERINSTRUCT's user and model control tokens.

We avoid the separation of `<\|im_start\|>` and `<\|im_end\|>` into multiple tokens by extending the tokenizer for TOWERINSTRUCT with two dedicated tokens. We do not explicitly add new tokens for user and assistant, as both strings already have dedicated tokens. Additionally, we overwrite the end-of-sequence token with the `<\|im_end\|>` token.

## F.2  Hyperparameters

Table 11 details the hyperparameter configuration for TOWERINSTRUCT's training. We also utilized bfloat16 mixed precision and packing.

| | |
|---|---|
| Global train batch size | 256 |
| Number of Epochs | 4 |
| Learning rate | 7e-6 |
| LR Scheduler | cosine |
| Warmup Steps | 500 |
| Weight Decay | 0.01 |
| Optimizer | Adam (Kingma & Ba, 2015) |
| Adam $\beta_1$ | 0.9 |
| Adam $\beta_2$ | 0.999 |
| Adam $\epsilon$ | 1e-8 |
| Maximum Sequence Length | 2048 |

Table 11: Hyperparameter configuration to finetune TOWERINSTRUCT on TOWERBLOCKS.

## G Translation full results

Tables 12, 13, 14 and 15 report translation quality on all test sets using various metrics: XCOMET, COMETKIWI-22, BLEURT, and CHRF. Tables 16, 17, 18 and 19 analyse translation quality for our languages, considering en→xx and xx→en translation directions, for the same metrics. On Tables 20, 21, and 22, we present translation results for a wider variety of models, broken down by language pair. Importantly, quality trends hold across metrics. TOWERINSTRUCT 13B is the open system with highest translation quality and is competitive with the closed model GPT-4.

| Models | FLORES-200 | | WMT 23 | | TICO 19 |
|---|---|---|---|---|---|
| | en→xx | xx→en | en→xx | xx→en | en→xx |
| **Closed** | | | | | |
| GPT-3.5-turbo | 94.41 2 | **95.54** 1 | 88.99 2 | 89.75 2 | 91.19 2 |
| GPT-4 | **94.75** 1 | **96.01** 1 | **89.46** 1 | **90.28** 1 | 91.38 2 |
| **Open** | | | | | |
| NLLB 54B | 90.04 4 | 93.78 4 | 78.99 6 | 81.38 6 | 90.11 3 |
| LLaMA-2 70B | 92.80 4 | 94.15 4 | 84.85 6 | 87.21 5 | 89.02 5 |
| Mixtral-8x7B-Instruct | 91.90 3 | 94.40 3 | 85.67 6 | 87.81 4 | 89.30 4 |
| ALMA-R 7B | — | — | 86.50 4 | 87.67 4 | — |
| ALMA-R 13B | — | — | 88.88 2 | 88.97 3 | — |
| TOWERINSTRUCT 7B | 93.85 2 | 94.67 3 | 87.20 4 | 87.88 4 | 90.56 3 |
| TOWERINSTRUCT 13B | 94.80 1 | 95.22 2 | 88.71 2 | 88.65 3 | 91.30 2 |

Table 12: Translation quality on WMT23 and TICO-19 by language pair measured by XCOMET. Models with statistically significant performance are grouped in quality clusters. Best performing models are in bold and best performing open models are underlined.

| Models | FLORES-200 | | WMT 23 | | TICO 19 |
|---|---|---|---|---|---|
| | en→xx | xx→en | en→xx | xx→en | en→xx |
| **Closed** | | | | | |
| GPT-3.5-turbo | 86.25 2 | 85.64 2 | 80.82 2 | 80.35 2 | 85.65 2 |
| GPT-4 | **86.42** 1 | **85.77** 1 | **81.20** 1 | **80.54** 1 | 85.79 2 |
| **Open** | | | | | |
| NLLB 54B | 82.93 5 | 84.89 4 | 70.96 6 | 76.69 5 | 85.16 3 |
| LLaMA-2 70B | 85.30 4 | 84.97 4 | 78.43 5 | 79.36 4 | 84.66 5 |
| Mixtral-8x7B-Instruct | 85.24 3 | 85.32 3 | 79.01 5 | 79.82 3 | 84.81 4 |
| ALMA-R 7B | — | — | 79.25 4 | 79.79 4 | — |
| ALMA-R 13B | — | — | 80.12 3 | 80.21 2 | — |
| TOWERINSTRUCT 7B | 85.96 3 | 85.41 3 | 79.80 4 | 79.95 3 | 85.32 3 |
| TOWERINSTRUCT 13B | 86.19 2 | 85.51 2 | 80.57 2 | 80.25 2 | 85.59 2 |

Table 13: Translation quality on WMT23 and TICO-19 by language pair measured by COMETKIWI-22. Models with statistically significant performance are grouped in quality clusters. Best performing models are in bold and best performing open models are underlined.

| Models | FLORES-200 | | WMT 23 | | TICO 19 |
|---|---|---|---|---|---|
| | en→xx | xx→en | en→xx | xx→en | en→xx |
| **Closed** | | | | | |
| GPT-3.5-turbo | **77.08** 1 | 78.12 3 | 72.06 2 | **72.50** 1 | 75.91 2 |
| GPT-4 | **77.26** 1 | 78.51 2 | **72.54** 1 | **72.91** 1 | 76.16 2 |
| **Open** | | | | | |
| NLLB 54B | 74.29 3 | 77.99 3 | 62.73 6 | 66.46 5 | 75.49 2 |
| LLaMA-2 70B | 75.04 4 | 78.28 2 | 68.03 5 | 71.01 3 | 74.00 4 |
| Mixtral-8x7B-Instruct | 74.78 3 | 78.10 2 | 68.81 5 | 71.32 3 | 74.22 4 |
| ALMA-R 7B | — | — | 68.64 5 | 70.66 4 | — |
| ALMA-R 13B | — | — | 70.09 4 | 71.47 3 | — |
| TOWERINSTRUCT 7B | 76.10 3 | 78.26 2 | 69.77 4 | 71.11 3 | 74.83 4 |
| TOWERINSTRUCT 13B | 76.89 2 | **78.67** 1 | 70.87 2 | 71.75 2 | 75.40 3 |

Table 14: Translation quality on WMT23 and TICO-19 by language pair measured by BLEURT. Models with statistically significant performance are grouped in quality clusters. Best performing models are in bold and best performing open models are underlined.

| Models | FLORES-200 | | WMT 23 | | TICO 19 |
|---|---|---|---|---|---|
| | en→xx | xx→en | en→xx | xx→en | en→xx |
| **Closed** | | | | | |
| GPT-3.5-turbo | **58.20** 1 | 63.75 3 | **56.38** 1 | 60.92 2 | 64.18 2 |
| GPT-4 | **58.61** 1 | 64.35 2 | **56.94** 1 | **61.33** 1 | 64.34 2 |
| **Open** | | | | | |
| NLLB 54B | 54.70 4 | 63.87 2 | 42.98 6 | 52.08 6 | 63.84 2 |
| LLaMA-2 70B | 55.19 4 | 64.15 2 | 52.31 4 | 59.66 2 | 61.65 4 |
| Mixtral-8x7B-Instruct | 54.50 4 | 63.38 3 | 51.22 4 | 58.63 4 | 61.34 4 |
| ALMA-R 7B | — | — | 45.20 7 | 57.33 4 | — |
| ALMA-R 13B | — | — | 46.52 6 | 58.37 3 | — |
| TOWERINSTRUCT 7B | 56.16 3 | 64.08 2 | 52.25 4 | 58.88 4 | 62.07 4 |
| TOWERINSTRUCT 13B | 57.19 2 | **64.79** 1 | 54.10 3 | 59.78 2 | 62.81 3 |

Table 15: Translation quality on WMT23 and TICO-19 by language pair measured by CHRF. Models with statistically significant performance are grouped in quality clusters. Best performing models are in bold and best performing open models are underlined.

| Models | FLORES-200 en→xx | xx→en | WMT 23 en→xx | xx→en | TICO 19 en→xx |
|---|---|---|---|---|---|
| **Closed** | | | | | |
| GPT-3.5-turbo | 94.41 [2] | **95.54** [1] | 88.99 [2] | 89.75 [2] | 91.19 [2] |
| GPT-4 | **94.75** [1] | **96.01** [1] | **89.46** [1] | **90.28** [1] | 91.38 [2] |
| **Open** | | | | | |
| NLLB 54B | 90.04 [4] | 93.78 [4] | 78.99 [6] | 81.38 [6] | 90.11 [3] |
| LLaMA-2 70B | 92.80 [4] | 94.15 [4] | 84.85 [6] | 87.21 [5] | 89.02 [5] |
| Mixtral-8x7B-Instruct | 91.90 [3] | 94.40 [3] | 85.67 [6] | 87.81 [4] | 89.30 [4] |
| ALMA-R 7B | — | — | 86.50 [4] | 87.67 [4] | — |
| ALMA-R 13B | — | — | 88.88 [2] | 88.97 [3] | — |
| TOWERINSTRUCT 7B | 93.85 [2] | 94.67 [3] | 87.20 [4] | 87.88 [4] | 90.56 [3] |
| TOWERINSTRUCT 13B | **94.80** [1] | 95.22 [2] | 88.71 [2] | 88.65 [3] | 91.30 [2] |

Table 16: Translation quality on FLORES-200 by language pair measured by xCOMET. Models with statistically significant performance are grouped in quality clusters. Best performing models are in bold and best performing open models are underlined.

| Models | FLORES-200 (en→xx) de | es | fr | it | ko | nl | pt | ru | zh |
|---|---|---|---|---|---|---|---|---|---|
| **Closed** | | | | | | | | | |
| GPT-3.5-turbo | 85.15 [2] | **87.04** [1] | **87.18** [1] | **87.47** [1] | 86.92 [3] | **86.88** [1] | 85.69 [2] | 85.58 [2] | 84.37 [2] |
| GPT-4 | **85.27** [1] | **87.07** [1] | **87.25** [1] | **87.51** [1] | **87.47** [1] | **86.90** [1] | 85.68 [2] | **85.99** [1] | **84.68** [1] |
| **Open** | | | | | | | | | |
| NLLB 54B | 82.59 [6] | 85.18 [4] | 85.23 [4] | 85.66 [4] | 86.11 [4] | 84.71 [4] | 83.45 [5] | 83.56 [4] | 69.88 [7] |
| LLaMA-2 70B | 84.19 [5] | 86.40 [3] | 86.68 [3] | 86.77 [3] | 85.46 [5] | 85.87 [3] | 84.57 [4] | 84.59 [3] | 83.13 [5] |
| Mixtral-8x7B-Instruct | 84.72 [3] | 86.74 [2] | 87.04 [2] | 87.18 [2] | 83.49 [6] | 85.95 [3] | 84.99 [3] | 84.78 [3] | 82.30 [6] |
| TOWERINSTRUCT 7B | 84.41 [4] | 86.77 [2] | 87.08 [2] | 87.31 [2] | 86.70 [3] | 86.48 [2] | 85.57 [2] | 85.50 [2] | 83.78 [4] |
| TOWERINSTRUCT 13B | 84.73 [3] | **86.94** [1] | **87.18** [1] | **87.45** [1] | 87.22 [2] | 86.60 [2] | **85.85** [1] | 85.68 [2] | 84.09 [3] |

| Models | FLORES-200 (xx→en) de | es | fr | it | ko | nl | pt | ru | zh |
|---|---|---|---|---|---|---|---|---|---|
| **Closed** | | | | | | | | | |
| GPT-3.5-turbo | 84.64 [2] | 86.27 [2] | **86.48** [1] | 86.84 [2] | 85.69 [2] | 86.18 [2] | **85.31** [1] | 84.59 [2] | 84.76 [2] |
| GPT-4 | **84.71** [1] | **86.39** [1] | **86.50** [1] | **86.95** [1] | **86.15** [1] | **86.25** [1] | **85.31** [1] | **84.75** [1] | **84.92** [1] |
| **Open** | | | | | | | | | |
| NLLB 54B | 84.09 [5] | 85.51 [5] | 86.04 [3] | 86.06 [4] | 85.13 [4] | 85.59 [5] | 84.45 [4] | 83.95 [4] | 83.18 [6] |
| LLaMA-2 70B | 84.29 [4] | 85.78 [4] | 86.05 [3] | 86.38 [3] | 84.45 [6] | 85.56 [5] | 84.87 [3] | 83.77 [4] | 83.57 [5] |
| Mixtral-8x7B-Instruct | 84.45 [3] | 86.07 [3] | 86.34 [2] | 86.78 [2] | 84.74 [5] | 85.78 [4] | 85.13 [2] | 84.45 [3] | 84.14 [4] |
| TOWERINSTRUCT 7B | 84.41 [3] | 86.12 [3] | 86.35 [2] | 86.79 [2] | 85.21 [4] | 85.98 [3] | 85.17 [2] | 84.47 [2] | 84.16 [4] |
| TOWERINSTRUCT 13B | 84.44 [3] | 86.09 [3] | 86.39 [2] | 86.83 [2] | 85.47 [3] | 86.04 [3] | 85.17 [2] | **84.69** [1] | 84.47 [3] |

Table 17: Translation quality on FLORES-200 by language pair measured by COMETKIWI-22. Models with statistically significant performance are grouped in quality clusters. Best performing models are in bold and best performing open models are underlined.

|  | \multicolumn{9}{c}{FLORES-200 (en→xx)} |
| Models | de | es | fr | it | ko | nl | pt | ru | zh |
| --- | --- | --- | --- | --- | --- | --- | --- | --- | --- |
| **Closed** | | | | | | | | | |
| GPT-3.5-turbo | **79.09**[1] | **76.75**[1] | **79.54**[1] | 79.83[2] | 69.39[2] | **77.79**[1] | **80.31**[1] | 77.31[2] | 73.69[2] |
| GPT-4 | **79.13**[1] | **76.64**[1] | **79.29**[1] | 80.00[2] | **70.31**[1] | 77.58[2] | **80.22**[1] | **78.16**[1] | **73.98**[1] |
| **Open** | | | | | | | | | |
| NLLB 54B | 77.71[3] | 75.37[4] | 77.96[3] | 79.26[3] | 68.95[2] | 76.47[3] | 77.80[4] | 76.81[3] | 58.32[6] |
| LLaMA-2 70B | 76.75[4] | 75.28[5] | 76.96[4] | 78.70[4] | 67.01[3] | 75.98[4] | 77.50[4] | 75.79[4] | 71.41[4] |
| Mixtral-8x7B-Instruct | 77.73[3] | 76.08[3] | 78.39[3] | 79.57[3] | 61.77[4] | 76.35[3] | 78.14[3] | 76.06[4] | 68.94[5] |
| TOWERINSTRUCT 7B | 77.61[3] | 75.71[4] | 78.03[3] | 79.58[3] | 69.25[2] | **77.73**[1] | 78.43[3] | 77.02[2] | 71.53[4] |
| TOWERINSTRUCT 13B | 78.15[2] | 76.42[2] | 78.96[2] | **80.39**[1] | **70.53**[1] | **77.93**[1] | 78.78[2] | **77.97**[1] | 72.85[3] |

|  | \multicolumn{9}{c}{FLORES-200 (xx→en)} |
| Models | de | es | fr | it | ko | nl | pt | ru | zh |
| --- | --- | --- | --- | --- | --- | --- | --- | --- | --- |
| **Closed** | | | | | | | | | |
| GPT-3.5-turbo | 80.38[2] | 77.27[3] | 80.55[3] | 77.91[3] | 75.22[3] | 77.02[2] | 80.86[3] | 77.73[3] | 76.12[2] |
| GPT-4 | **80.74**[1] | 77.61[2] | 80.72[2] | 78.14[2] | **76.51**[1] | **77.23**[1] | 81.11[2] | 78.02[2] | **76.54**[1] |
| **Open** | | | | | | | | | |
| NLLB 54B | 80.12[3] | 77.09[3] | 80.64[2] | 77.79[3] | 75.32[2] | 76.99[2] | 80.81[3] | 77.95[2] | 75.19[4] |
| LLaMA-2 70B | 80.38[2] | 77.65[1] | 80.79[2] | 78.05[2] | 75.58[2] | 76.77[3] | 81.16[2] | 78.18[2] | 75.96[2] |
| Mixtral-8x7B-Instruct | 80.40[2] | 77.79[1] | 80.75[2] | 78.53[1] | 74.15[4] | 76.87[2] | 80.85[3] | 78.02[2] | 75.57[3] |
| TOWERINSTRUCT 7B | 80.17[3] | 77.47[2] | 80.67[2] | 78.40[1] | 75.62[2] | 76.96[2] | 81.30[2] | 78.10[2] | 75.68[3] |
| TOWERINSTRUCT 13B | **80.55**[1] | **77.65**[1] | **81.03**[1] | **78.54**[1] | **76.53**[1] | **77.22**[1] | **81.51**[1] | **78.51**[1] | **76.46**[1] |

Table 18: Translation quality on FLORES-200 by language pair measured by BLEURT. Models with statistically significant performance are grouped in quality clusters. Best performing models are in bold and best performing open models are underlined.

| Models | FLORES-200 (en→xx) | | | | | | | | |
|---|---|---|---|---|---|---|---|---|---|
| | de | es | fr | it | ko | nl | pt | ru | zh |
| **Closed** | | | | | | | | | |
| GPT-3.5-turbo | 67.22 [2] | **57.39** [1] | **72.79** [1] | **60.67** [1] | 35.49 [2] | 59.57 [2] | **72.96** [1] | 58.48 [2] | **39.21** [1] |
| GPT-4 | **67.89** [1] | 57.13 [2] | **72.89** [1] | **60.60** [1] | **37.18** [1] | **59.97** [1] | 72.98 [1] | **59.50** [1] | **39.32** [1] |
| **Open** | | | | | | | | | |
| NLLB 54B | 63.18 [5] | 55.30 [5] | 70.25 [3] | 58.83 [3] | **36.54** [1] | 56.99 [5] | 68.19 [4] | 57.28 [3] | 25.73 [5] |
| LLaMA-2 70B | 63.43 [5] | 55.39 [5] | 69.54 [4] | 58.20 [3] | 32.07 [3] | 56.53 [5] | _69.61_ [2] | 56.58 [4] | 35.38 [3] |
| Mixtral-8x7B-Instruct | 64.14 [4] | 56.14 [4] | _70.91_ [2] | 59.01 [2] | 27.54 [4] | 56.22 [6] | _69.43_ [2] | 56.07 [4] | 31.01 [4] |
| TOWERINSTRUCT 7B | 63.87 [4] | 56.04 [4] | 70.23 [3] | 59.45 [2] | 35.44 [2] | 58.16 [4] | 68.74 [4] | 57.77 [3] | 35.78 [3] |
| TOWERINSTRUCT 13B | _65.16_ [3] | _56.58_ [3] | _71.26_ [2] | **60.32** [1] | **37.10** [1] | _59.04_ [3] | 69.06 [3] | _58.77_ [2] | _37.40_ [2] |

| Models | FLORES-200 (xx→en) | | | | | | | | |
|---|---|---|---|---|---|---|---|---|---|
| | de | es | fr | it | ko | nl | pt | ru | zh |
| **Closed** | | | | | | | | | |
| GPT-3.5-turbo | 69.31 [2] | 60.46 [3] | 69.54 [2] | 62.76 [3] | 57.50 [3] | 60.75 [2] | 72.56 [3] | 62.80 [3] | 58.07 [2] |
| GPT-4 | **69.74** [1] | 61.09 [2] | **69.94** [1] | 62.75 [3] | **59.55** [1] | 60.88 [2] | 72.91 [3] | 63.40 [2] | **58.87** [1] |
| **Open** | | | | | | | | | |
| NLLB 54B | 68.54 [3] | 60.72 [2] | 69.70 [2] | 62.95 [3] | 58.55 [2] | 60.67 [2] | 72.26 [3] | 62.66 [3] | _58.83_ [1] |
| LLaMA-2 70B | 69.22 [2] | **61.34** [1] | **70.08** [1] | 63.51 [2] | 57.82 [2] | 60.90 [2] | 72.96 [2] | 63.61 [2] | 57.94 [2] |
| Mixtral-8x7B-Instruct | 69.00 [2] | **61.29** [1] | 69.32 [2] | 63.38 [2] | 55.56 [4] | 59.98 [3] | 72.18 [4] | 62.77 [3] | 56.97 [3] |
| TOWERINSTRUCT 7B | 68.94 [2] | **61.39** [1] | 69.56 [2] | 63.59 [2] | 58.48 [2] | 60.65 [2] | 73.00 [2] | 63.37 [2] | 57.79 [2] |
| TOWERINSTRUCT 13B | **69.39** [1] | **61.50** [1] | **70.07** [1] | **64.06** [1] | **59.81** [1] | **61.40** [1] | **73.54** [1] | **64.41** [1] | **58.90** [1] |

Table 19: Translation quality on FLORES-200 by language pair measured by CHRF. Models with statistically significant performance are grouped in quality clusters. Best performing models are in bold and best performing open models are underlined.

| Models | de | es | fr | FLORES-200 (en→xx) it | ko | nl | pt | ru | zh |
|---|---|---|---|---|---|---|---|---|---|
| **Closed** | | | | | | | | | |
| GPT-3.5-turbo | 88.78 | 87.08 | 89.02 | 89.06 | 89.36 | 88.63 | 90.46 | 89.56 | 88.58 |
| GPT-4 | 88.98 | 87.10 | 88.93 | 89.05 | 90.06 | 88.56 | 90.43 | 90.19 | 88.87 |
| **Open** | | | | | | | | | |
| NLLB 54B | 87.18 | 85.92 | 87.71 | 88.10 | 89.00 | 87.33 | 88.72 | 88.89 | 78.26 |
| LLaMA-2 7B | 84.03 | 84.37 | 85.18 | 85.18 | 80.20 | 84.48 | 87.01 | 85.09 | 82.50 |
| LLaMA-2 13B | 85.60 | 85.45 | 86.74 | 87.02 | 84.22 | 86.11 | 88.33 | 87.02 | 84.83 |
| LLaMA-2 70B | 87.31 | 86.41 | 87.82 | 88.22 | 88.07 | 87.47 | 89.11 | 88.65 | 87.32 |
| Mistral-7B-Instruct-v0.2 | 84.27 | 84.87 | 86.16 | 85.86 | 79.20 | 84.43 | 87.53 | 85.78 | 82.41 |
| Mixtral-8x7B | 87.95 | 86.64 | 88.39 | 88.44 | 85.72 | 87.26 | 89.34 | 88.89 | 86.23 |
| Mixtral-8x7B-Instruct | 87.99 | 86.80 | 88.53 | 88.77 | 85.63 | 87.57 | 89.45 | 89.09 | 85.99 |
| Qwen1.5 72B | 87.20 | 86.46 | 87.78 | 88.19 | 87.64 | 87.40 | 89.13 | 88.41 | 88.85 |
| Gemma 7B | 86.13 | 85.84 | 87.09 | 87.03 | 84.89 | 86.03 | 88.60 | 87.24 | 85.75 |
| COMMAND R | 87.60 | 86.95 | 87.90 | 88.93 | 89.19 | 87.80 | 89.74 | 89.55 | 88.30 |
| COMMAND R+ | 88.87 | 87.45 | 89.13 | 89.46 | 90.25 | 88.67 | 90.44 | 90.57 | 89.18 |
| ALMA-PRETRAIN 7B | 86.47 | 83.18 | 84.23 | 83.59 | 68.06 | 81.05 | 84.80 | 87.96 | 85.80 |
| ALMA-PRETRAIN 13B | 87.07 | 84.90 | 86.05 | 86.09 | 77.10 | 84.36 | 87.47 | 88.91 | 86.58 |
| **TOWER** | | | | | | | | | |
| TOWERBASE 7B | 86.91 | 85.95 | 87.76 | 87.93 | 86.55 | 87.37 | 89.47 | 88.72 | 86.48 |
| TOWERBASE 13B | 87.21 | 86.01 | 88.34 | 88.25 | 88.78 | 87.52 | 89.36 | 88.30 | 87.14 |
| TOWERINSTRUCT 7B | 87.82 | 86.76 | 88.44 | 88.73 | 89.41 | 88.38 | 89.60 | 89.53 | 87.90 |
| TOWERINSTRUCT 13B | 88.16 | 87.06 | 88.92 | 89.21 | 89.92 | 88.63 | 89.78 | 89.95 | 88.29 |

| Models | de | es | fr | FLORES-200 (xx→en) it | ko | nl | pt | ru | zh |
|---|---|---|---|---|---|---|---|---|---|
| **Closed** | | | | | | | | | |
| GPT-3.5-turbo | 89.60 | 87.26 | 89.46 | 88.03 | 87.83 | 87.71 | 89.78 | 86.69 | 86.92 |
| GPT-4 | 89.76 | 87.57 | 89.61 | 88.21 | 88.58 | 87.88 | 89.94 | 86.94 | 87.29 |
| **Open** | | | | | | | | | |
| NLLB 54B | 89.17 | 87.25 | 89.29 | 87.91 | 87.86 | 87.49 | 89.38 | 86.66 | 86.55 |
| LLaMA-2 7B | 88.47 | 86.63 | 88.78 | 87.48 | 85.52 | 86.67 | 88.98 | 85.87 | 85.53 |
| LLaMA-2 13B | 89.01 | 86.98 | 89.14 | 87.87 | 86.95 | 87.23 | 89.26 | 86.37 | 86.35 |
| LLaMA-2 70B | 89.44 | 87.49 | 89.55 | 88.18 | 87.91 | 87.52 | 89.84 | 86.87 | 86.91 |
| Mistral-7B-Instruct-v0.2 | 88.83 | 87.07 | 88.81 | 87.69 | 85.16 | 86.93 | 89.05 | 86.21 | 85.65 |
| Mixtral-8x7B | 89.55 | 87.57 | 89.58 | 88.35 | 87.03 | 87.54 | 89.80 | 86.79 | 86.63 |
| Mixtral-8x7B-Instruct | 89.57 | 87.65 | 89.56 | 88.44 | 87.37 | 87.54 | 89.73 | 86.81 | 86.88 |
| Qwen1.5 72B | 89.67 | 87.66 | 89.58 | 88.41 | 88.42 | 87.72 | 89.88 | 87.13 | 87.94 |
| Gemma 7B | 89.17 | 87.09 | 89.12 | 87.81 | 87.28 | 87.23 | 89.48 | 86.59 | 86.59 |
| COMMAND R | 89.15 | 87.51 | 88.91 | 88.05 | 88.29 | 87.49 | 89.47 | 86.65 | 87.10 |
| COMMAND R+ | 89.07 | 87.87 | 89.67 | 88.53 | 88.82 | 87.10 | 90.15 | 87.32 | 87.91 |
| ALMA-PRETRAIN 7B | 89.23 | 86.84 | 89.01 | 87.68 | 83.35 | 86.92 | 89.05 | 86.81 | 86.59 |
| ALMA-PRETRAIN 13B | 89.81 | 87.42 | 89.42 | 88.18 | 86.26 | 87.59 | 89.70 | 87.23 | 87.16 |
| **TOWER** | | | | | | | | | |
| TOWERBASE 7B | 89.26 | 87.15 | 89.47 | 88.14 | 87.80 | 87.45 | 89.77 | 86.41 | 86.72 |
| TOWERBASE 13B | 89.54 | 87.42 | 89.55 | 88.11 | 88.24 | 87.61 | 89.71 | 86.18 | 87.02 |
| TOWERINSTRUCT 7B | 89.48 | 87.48 | 89.50 | 88.39 | 88.16 | 87.66 | 89.92 | 86.90 | 86.96 |
| TOWERINSTRUCT 13B | 89.61 | 87.62 | 89.67 | 88.42 | 88.48 | 87.92 | 90.07 | 87.20 | 87.27 |

Table 20: COMET-22 on FLORES-200 for a wide variety of models.

| Models | WMT23 | | | | | |
|---|---|---|---|---|---|---|
| | en→de | en→ru | en→zh | de→en | ru→en | zh→en |
| **Closed** | | | | | | |
| GPT-3.5-turbo | 84.61 | 85.38 | 86.70 | 85.91 | 83.02 | 81.52 |
| GPT-4 | 84.89 | 86.07 | 87.08 | 86.17 | 83.63 | 81.27 |
| **Open** | | | | | | |
| NLLB 54B | 77.40 | 83.91 | 74.48 | 80.06 | 80.52 | 76.60 |
| LLaMA-2 7B | 75.02 | 77.87 | 79.16 | 83.36 | 80.58 | 77.40 |
| LLaMA-2 13B | 78.29 | 80.44 | 81.30 | 83.92 | 81.54 | 78.73 |
| LLaMA-2 70B | 81.62 | 83.04 | 84.19 | 85.12 | 82.84 | 79.73 |
| Mistral-7B-Instruct-v0.2 | 76.78 | 80.27 | 81.26 | 84.18 | 81.52 | 79.11 |
| Mixtral-8x7B | 81.92 | 83.39 | 83.81 | 85.04 | 82.70 | 79.50 |
| Mixtral-8x7B-Instruct | 83.07 | 83.79 | 83.94 | 85.45 | 83.02 | 80.04 |
| Qwen1.5 72B | 81.44 | 83.31 | 86.48 | 85.54 | 83.01 | 80.60 |
| Gemma 7B | 79.56 | 82.20 | 83.56 | 84.60 | 82.14 | 79.24 |
| COMMAND R | 83.29 | 84.96 | 84.92 | 84.66 | 82.29 | 79.61 |
| COMMAND R+ | 84.85 | 86.22 | 86.31 | 84.82 | 83.38 | 80.59 |
| ALMA-PRETRAIN 7B | 80.20 | 83.01 | 82.68 | 83.51 | 81.82 | 78.66 |
| ALMA-PRETRAIN 13B | 81.18 | 83.72 | 83.83 | 84.32 | 82.71 | 79.22 |
| ALMA-R 7B | 82.41 | 84.28 | 83.51 | 84.55 | 82.50 | 80.13 |
| ALMA-R 13B | 83.59 | 85.37 | 84.43 | 85.39 | 83.23 | 80.48 |
| **TOWER** | | | | | | |
| TOWERBASE 7B | 81.03 | 83.25 | 84.00 | 84.09 | 80.08 | 78.92 |
| TOWERBASE 13B | 81.18 | 83.46 | 84.03 | 83.89 | 80.03 | 78.94 |
| TOWERINSTRUCT 7B | 83.22 | 84.73 | 84.89 | 85.24 | 82.94 | 80.13 |
| TOWERINSTRUCT 13B | 83.98 | 85.51 | 85.92 | 85.62 | 83.21 | 80.72 |

Table 21: COMET-22 on WMT23 for a wide variety of models.

| Models | WMT23 | | | | |
|---|---|---|---|---|---|
| | en→es | en→fr | en→pt | en→ru | en→zh |
| **Closed** | | | | | |
| GPT-3.5-turbo | 88.67 | 81.86 | 90.30 | 87.88 | 88.09 |
| GPT-4 | 88.76 | 81.85 | 90.30 | 88.36 | 88.32 |
| **Open** | | | | | |
| NLLB 54B | 88.74 | 82.01 | 89.84 | 88.67 | 85.97 |
| LLaMA-2 7B | 85.77 | 78.08 | 86.97 | 82.99 | 81.86 |
| LLaMA-2 13B | 86.94 | 79.83 | 88.48 | 85.44 | 84.89 |
| LLaMA-2 70B | 87.84 | 80.67 | 89.24 | 87.12 | 87.44 |
| Mistral-7B-Instruct-v0.2 | 86.25 | 79.18 | 87.87 | 84.35 | 84.13 |
| Mixtral-8x7B | 88.12 | 81.15 | 89.27 | 87.14 | 86.58 |
| Mixtral-8x7B-Instruct | 88.23 | 81.39 | 89.48 | 87.04 | 86.84 |
| Qwen1.5 72B | 86.08 | 80.32 | 88.20 | 80.53 | 86.68 |
| Gemma 7B | 87.30 | 78.20 | 88.66 | 86.16 | 86.78 |
| COMMAND R | 88.35 | 81.41 | 89.62 | 88.65 | 86.43 |
| COMMAND R+ | 88.99 | 82.07 | 90.30 | 89.06 | 88.28 |
| ALMA-PRETRAIN 7B | 84.42 | 76.74 | 84.92 | 86.53 | 85.27 |
| ALMA-PRETRAIN 13B | 86.17 | 79.09 | 87.56 | 87.27 | 86.54 |
| ALMA-R 7B | 84.63 | 76.02 | 82.92 | 87.80 | 85.41 |
| ALMA-R 13B | 85.93 | 79.90 | 87.41 | 88.58 | 86.22 |
| **TOWER** | | | | | |
| TOWERBASE 7B | 87.90 | 81.20 | 89.45 | 86.94 | 86.97 |
| TOWERBASE 13B | 87.90 | 81.48 | 89.54 | 87.26 | 87.57 |
| TOWERINSTRUCT 7B | 88.34 | 81.60 | 89.38 | 88.11 | 87.63 |
| TOWERINSTRUCT 13B | 88.63 | 81.82 | 89.48 | 88.49 | 88.20 |

Table 22: COMET-22 on TICO-19 for a wide variety of models.

# H    Translation-related tasks full results

## H.1    Languages considered

For APE, on Table 3, we consider 4 language pairs: en→de, en→zh, de→en, and ru→en. We leave out en→ru and zh→en, because we had no post editions to serve as fewshot examples for LLaMA-2 and Mixtral-8x7B-Instruct. Nevertheless, we provide results for TOWERINSTRUCT, GPT-3.5-turbo, and GPT-4 on the 6 language pairs in Table 23.

For NER, we consider English, German, French, Spanish, Italian, Portuguese, Russian, and Chinese. Finally, we evaluate GEC on English, German, and Spanish. For this task, besides the results in Table 3, we also measure ERRANT in Table 24.

Results broken down by language are in Tables 25, 26, and 27.

| | APE | |
| Models | en→xx | xx→en |
|---|---|---|
| Baseline (no edits) | 78.84 4 | 78.80 4 |
| GPT-3.5-turbo | 82.32 3 | 77.91 5 |
| GPT-4 | **85.52 1** | **83.12 1** |
| TOWERINSTRUCT 7B | 83.10 3 | 80.19 3 |
| TOWERINSTRUCT 13B | 83.65 2 | 80.89 2 |

Table 23: APE results for the 6 WMT23 LPs considered. NLLB corresponds to the translations that were subject to editing, so their quality serves as the baseline for the task. Table 3 did not include zh-en and en-ru to guarantee a fair comparison with open models — there were no fewshot examples available for these LPs.

| | GEC |
| Models | Multilingual |
|---|---|
| **Closed** | |
| GPT-3.5-turbo | **0.49 1** |
| GPT-4 | 0.48 3 |
| **Open** | |
| LLaMA-2 70B | 0.43 4 |
| Mixtral-8x7B-Instruct | 0.43 4 |
| TOWERINSTRUCT 7B | 0.42 4 |
| TOWERINSTRUCT 13B | 0.43 4 |

Table 24: GEC ERRANT results.

| Models | WMT23 en→de | en→ru | en→zh | de→en | ru→en | zh→en |
|---|---|---|---|---|---|---|
| Baseline (no edits) | 77.87 | 82.93 | 75.72 | 79.92 | 80.05 | 76.44 |
| **Closed** | | | | | | |
| GPT-3.5-turbo | 80.67 | 84.03 | 82.27 | 78.48 | 78.88 | 76.37 |
| GPT-4 | 84.65 | 86.15 | 85.75 | 85.39 | 83.21 | 80.75 |
| **Open** | | | | | | |
| GPT-3.5-turbo | 80.67 | 84.03 | 82.27 | 78.48 | 78.88 | 76.37 |
| GPT-4 | 84.65 | 86.15 | 85.75 | 85.39 | 83.21 | 80.75 |
| LLaMA-2 70B | 78.49 | — | 78.20 | 81.30 | 80.76 | — |
| Mixtral-8x7B-Instruct | 82.12 | — | 83.15 | 83.40 | 82.22 | — |
| **TOWER** | | | | | | |
| TOWERINSTRUCT 7B | 81.86 | 83.92 | 83.52 | 82.29 | 80.82 | 77.45 |
| TOWERINSTRUCT 13B | 82.03 | 84.34 | 84.59 | 83.22 | 81.30 | 78.15 |

Table 25: APE COMET-22 results by language pair.

| Models | en | de | es |
|---|---|---|---|
| Baseline (no edits) | 13.75 | 18.23 | 18.00 |
| **Closed** | | | |
| GPT-3.5-turbo | 14.71 | 13.19 | 17.29 |
| GPT-4 | 16.48 | 12.89 | 15.86 |
| **Open** | | | |
| LLaMA-2 70B | 17.46 | 20.67 | 27.09 |
| Mixtral-8x7B-Instruct | 16.44 | 15.38 | 19.47 |
| **TOWER** | | | |
| TOWERINSTRUCT 7B | 13.39 | 14.77 | 17.23 |
| TOWERINSTRUCT 13B | 13.13 | 14.42 | 19.48 |

Table 26: GEC edit rate results by language.

| Models | en | de | es | fr | it | pt | zh |
|---|---|---|---|---|---|---|---|
| **Closed** | | | | | | | |
| GPT-3.5-turbo | 55.43 | 60.12 | 56.82 | 53.34 | 55.46 | 52.57 | 17.82 |
| GPT-4 | 63.61 | 66.58 | 65.24 | 58.72 | 63.39 | 61.74 | 39.88 |
| **Open** | | | | | | | |
| LLaMA-2 70B | 46.34 | 48.79 | 50.69 | 47.50 | 53.96 | 45.60 | 19.44 |
| Mixtral-8x7B-Instruct | 45.74 | 46.94 | 46.03 | 46.11 | 50.86 | 40.21 | 16.51 |
| **TOWER** | | | | | | | |
| TOWERINSTRUCT 7B | 75.09 | 78.01 | 74.89 | 70.35 | 76.39 | 73.88 | 53.13 |
| TOWERINSTRUCT 13B | 77.52 | 79.73 | 76.69 | 74.55 | 80.36 | 77.47 | 56.57 |

Table 27: NER F1 results by language.

