# OpenReview forum: "Tower: An Open Multilingual Large Language Model for Translation-Related Tasks"
_colmweb.org/COLM/2024/Conference — COLM_

### Official Review · Reviewer_E4m3 · 2024-04-15

**Rating:** 9
**Confidence:** 4
**Ethics Flag:** 1

**Summary:**

The authors describe the process of adapting large language models to translation-related tasks (e.g., translation, pre-editing, post-editing) in great detail. They evaluate the resulting models (TOWER-BASE and TOWER-INSTRUCT) against open and closed models on a variety of tasks, showing clear performance improvements (outperforming open models and approaching or on par with large closed models). The authors pledge to release their models, code, and data sets to the community.

**Questions To Authors:**

Appendix C.2 states that the parallel data used in continued pre-training is prompted with "<SRC LANG>: <SRC>\n<TGT LANG>: <TGT>". This is an important enough design decision to move to the main paper (e.g., that parallel data is labeled/prompted instead of simply concatenated).

**Reasons To Accept:**

This work contains two types of contributions that can significantly benefit the field:
- Methods, results, and observations related to effectively customizing existing LLMs to a class of tasks. These can be directly used or extended in ongoing work.
- Actual models, code, and data. These can be deployed to benefit real-world users as well as used as starting points for further exploration.

On a more detailed level:
- The authors thoroughly describe the processes used to construct data sets, train models, and run evaluations.
- The authors include detailed analyses (including ablation experiments) to explore where and how different techniques are effective.
- The authors include a variety of sound baselines that make the results credible.

**Reasons To Reject:**

Since the authors' models are based on Llama 2, they inherit Llama's usage restrictions (cannot be used to improve non-Llama models and cannot be used by some companies: https://llama.meta.com/llama-downloads/). This could make it challenging for some members of the research community to build on this work.

---

> ### Author Rebuttal · Authors · 2024-05-30
>
> Thank you for your positive review and insightful comments! We are happy to hear that you found our contributions can significantly benefit the field.
>
> > Since the authors' models are based on Llama 2, they inherit Llama's usage restrictions (cannot be used to improve non-Llama models and cannot be used by some companies: https://llama.meta.com/llama-downloads/). This could make it challenging for some members of the research community to build on this work.
>
> This is an important point. Our Tower recipe can be used with other open models, which are currently launched at a fast pace. In fact, we are planning to release Tower models based on Mistral-7B, which will allow us to reach a wider part of the community.
>
> > Appendix C.2 states that the parallel data used in continued pre-training is prompted with “<SRC LANG>: <SRC>\n<TGT LANG>: <TGT>”. This is an important enough design decision to move to the main paper (e.g., that parallel data is labeled/prompted instead of simply concatenated).
>
> Thank you for pointing this out. We agree that this is an important design decision given our focus on parallel data during continued pretraining. We will add this information to the main text in the final version.

---

> > ### Comment · Reviewer_E4m3 · 2024-06-03
> > **Re: Rebuttal by Authors**
> >
> > Thank you for adding this information. The first point is strong enough to mention in the abstract and introduction: beyond training and releasing specific models, you share a recipe that can create Tower-style models from newer and stronger open models as they are available.

---

### Official Review · Reviewer_Ls3C · 2024-04-24

**Rating:** 8
**Confidence:** 4
**Ethics Flag:** 1

**Summary:**

This paper sets out to create a large language model specifically designed for translation and translation-related tasks. This is done by continual pre-training of LLaMA-2 on monolingual and parallel data in multiple languages, then instruction-tuning on a curated and filtered labeled dataset of translation-related tasks (translation, post-editing, named entity recognition).

The resulting model, TowerInstruct, achieves impressive results on translation, named entity recognition, automatic post-editing, and grammatical error correction (with GEC being a held-out task unseen during instruction-tuning), performing on par with much larger models. Furthermore, the authors publicly release the model, dataset, and evaluations.

This is an important contribution that will be of interest to the industry. My main quibble is that I wish that the set of translation tasks explored were more extensive (e.g. including different domains or modalities).

**Questions To Authors:**

1. Not a question, but a suggestion: "automatic post-edition" should be "automatic post-editing" (in a couple of places).

2. Why did you select those 10 languages? (EN, DE, FR, NL, IT, ES, PT, KO, RU, ZH)

3. L63: Why do you need language identification?

**Reasons To Accept:**

1. The approach to training a translation-focused LLM is useful. Furthermore, it is effective, as shown by several extensive evaluations.

2. In addition to the paper, the authors will release their LLM, dataset, and evaluation framework. This will foster future research in the area.

3. The analysis in section 4 is interesting and clearly shows the impact of each component of the training "recipe".

**Reasons To Reject:**

1. I am a bit perplexed by the choice of "translation-related" tasks in the paper: translation, automatic post-editing, named entity recognition, and grammatical error correction. I don't think named entity recognition or grammatical error correction are specifically translation-related (although they may be useful for translation, like several other tasks), while more related tasks are omitted (e.g., metric development, quality estimation, parallel data curation, bilingual lexicon induction). In addition, I think it would be useful to see more comprehensive coverage of machine translation -- including subtasks like domain-specific MT, low-resource or dialectal translation, or multimodal MT.

2. In some cases, it is unclear how experimental settings are selected. Examples: L100 "We train for 4 epochs using a low learning rate and a large batch size", L121 "We prompt TOWER and closed models in a 0-shot fashion and others with 5 examples randomly selected from the development set." It would be good to know how these design decisions were made (optimization on a development set?).

---

> ### Author Rebuttal · Authors · 2024-05-30
>
> Thank you for your positive reviews and suggestions!
>
> > I don't think named entity recognition or grammatical error correction are specifically translation-related […], while more related tasks are omitted […]
>
> NER and GEC are an important part of a real-world translation pipeline which prompted their inclusion.
>
> Yet, tasks such as MT evaluation are also important. We'll run these experiments for the final version. Preliminary results suggest that models trained with Tower’s recipe can be competitive with widespread metrics like COMET-22.
>
> > […] more comprehensive coverage of machine translation
>
> For this paper, we focused on widely used benchmarks covering different domains and languages: Flores (Wikipedia), WMT23 (news, e-commerce, speech, social, manuals), and TICO (medical). We performed a domain analysis but found no observable pattern in translation quality. We'll include this in the final version.
>
> Still, we acknowledge that exploring low-resource and multimodal MT are exciting avenues for future work.
>
> > […] unclear how experimental settings are selected.
>
> Training: we followed configurations from other instruction tuned models [1,2,3] and monitored perplexity. Most led to step-wise losses that have been observed with other LLMs, with sudden downward loss shifts at new epochs. Our configuration alleviated these issues. We'll clarify this.
>
> Evaluation: we evaluated pretrained models in a 5-shot fashion and instruction-following models in a 0-shot setting. The exception was Mixtral-Instruct, where a 5-shot prompt corrected output format issues found in responses in a 0-shot setting. For GEC, we evaluated all models in a 5-shot setting as it was a held-out task.
>
> > Why did you select those 10 languages?
>
> We aimed to cover a range of high-resource languages across multiple scripts, for which there is a lot of monolingual, parallel, and task-specific data. As such, our project differs from “massively multilingual MT” projects which prioritize language coverage. Extending Tower with more languages is an interesting direction for future research.
>
> > […] Why do you need language identification?
>
> Even on a large corpus of web texts such as mC4, there can be texts of other languages in language-specific partitions. By performing language identification, we further ensured we were training on the correct language.
>
> [1] https://arxiv.org/abs/2310.16944
>
> [2] https://huggingface.co/Intel/neural-chat-7b-v3-1
>
> [3] https://huggingface.co/teknium/OpenHermes-7B

---

### Official Review · Reviewer_7egk · 2024-05-14

**Rating:** 7
**Confidence:** 4
**Ethics Flag:** 1

**Summary:**

This paper discusses a multilingual recipe for large language models (LLMs) to have the LLMs acquire multilingual capability in the translation task and multilingual relevant tasks. In the recipe, the authors takes LLaMa-2 models as their open LLM models, perform continued pre-training in the LLM on a multilingual mixture of monolingual and parallel data, creating TOWERBASE, and do finetuning on instructions for translation processes, creating TOWERINSTRUCT. The experiments in the translation-related tasks show that the proposed  model outperforms open alternative on the tasks and achieves competitive performance wit closed models such as GPT-4.

**Questions To Authors:**

- The authors described the parallel-data training as "During training, loss is calculated on both source and target." in the 67th line. Can you clarify this by referring the prompt template (reported in C.2)? Did you calculate the loss over the whole template including <SRC-LANG> and/or <TGT_LANG>?

- The column of GEC in Table 3 seems wrong. Since it reports the error rate, I guess that lower score is better, hence the GPT-3.5-turbo should be best-ranked with the error rate of 15.06.

- Why did you use COMET-kiwi for data cleaning?

- What kind of prompts did you use in each of LLM baselines?

- Did you do full fine-tuning in all the experiments? Have you tried parameter efficient model finetuning like LoRA finetuning [1]?

[1] "LoRA: Low-Rank Adaptation of Large Language Models", Edward J. Hu, Yelong Shen, Phillip Wallis, Zeyuan Allen-Zhu, Yuanzhi Li, Shean Wang, Lu Wang, Weizhu Chen. arXiv:2106.09685 (cs).

**Reasons To Accept:**

The paper is well organized and easy to follow each section. Most parts are clearly described. Each result table is carefully designed, as those model ranking report helps understand.

The authors conducted a variety of experiments in the translation-related tasks. Their experimental results are technically sound, showing that the multilingual capability is well acquired by the proposed approach. The proposed approaches work well at different LLM model scale (7B->13B).

**Reasons To Reject:**

- Baselines seems weak as they evaluate (non-multilingual) models in multilingual tasks. Also, the system comparison against the previous work (Xu et al., 2024b) seems misleading and needs more clarification. The ALMA models are same LLaMa-based models but trained on different data sets, regardless of its different finetuning recipe. It is not clear to me whether those improvements come from either data set and/or the proposed TOWER recipe.

- The proposed approach is only applied to LLaMa-2 7B and 13B. We would love to know how much performance we would get from other open LLMs like Mistral-7B-Instruct.

---

> ### Author Rebuttal · Authors · 2024-05-30
>
> Thank you for your review!
>
> > Baselines seems weak […]
>
> Note that we compare with much larger open models trained on multilingual data, and SOTA closed LLMs which have been shown to be strong in multilingual tasks. For MT, we also compare with strong dedicated models.
>
> Still, we'll update the paper with results from newly released multilingual LLMs (e.g., Command R). Our experiments for MT suggest that TowerInstruct-13B outperforms Command R (35B), and is competitive with Command R+ (104B).
>
> > […] comparison against the previous work (Xu et al., 2024b) […] needs more clarification.
>
> Tower differs from ALMA in both continued pretraining (CPT) and supervised finetuning (SFT). During CPT, Tower mixes monolingual and parallel data — a key contribution of our paper — while ALMA uses only monolingual data. Our ablations show that the former outperforms the latter (Fig. 7).
>
> For SFT, Tower includes data from multiple tasks, while ALMA focuses only on MT. Combining all tasks yields a model with similar translation quality that can perform other tasks (Table 4).
>
> > […] how much performance we would get from other open LLMs like Mistral-7B-Instruct.
>
> We have now trained Mistral-7B with Tower’s recipe. This model outperforms TowerInstruct-7B across the board, and is on par with TowerInstruct-13B on MT while slightly underperforming it on translation-related tasks. We'll add this experiment in the final version.
>
> > […] Did you calculate the loss over the whole template […]
>
> We calculate the loss for parallel data during CPT on the whole template, including <SRC-LANG>, <TGT_LANG>, source, and target.
>
> > The column of GEC in Table 3 seems wrong.
>
> This is a valid point, and we alert readers to this possibility in L130: "a first cluster will not exist if no model significantly outperforms all others on a majority of languages". Better average performance may not always imply the highest rank.
>
> > Why […] COMET-kiwi for data cleaning?
>
> Filtering MT data with QE metrics has been shown to improve translation quality [1]. We'll clarify this and include the citation in the final version.
>
> > What kind of prompts did you use […]
>
> We followed prompts from literature when available. We'll specify them in the appendix and release a dataset with all prompts upon acceptance.
>
> > Have you tried […] LoRA […]
>
> Preliminary experiments indicated a drop in performance with LoRA, which led us to adopt full finetuning for all our experiments.
>
> [1] https://aclanthology.org/2023.wmt-1.50/

---

> > ### Comment · Reviewer_7egk · 2024-06-05
> > **Thank you for the response.**
> >
> > The response clarified my questions. The final ver will be more technically sound with those additional results. Hence, I increased the rating to "7: Good paper, accept".

---

### Official Review · Reviewer_i13c · 2024-05-20

**Rating:** 6
**Confidence:** 4
**Ethics Flag:** 1

**Summary:**

This paper discusses a method for improving the performance of LLMs in translation workflows by tailoring them to handle multiple tasks. The authors introduce TowerBase, created through continued pretraining on a multilingual mixture of monolingual and parallel data, and TowerInstruct, obtained by finetuning on instructions pertinent to translation processes. Their approach results in models that outperform other open LLM alternatives on various tasks and are competitive with general-purpose closed LLMs.

**Reasons To Accept:**

The paper explores continued pretraining and fine-tuning specifically for translation tasks. The methodology is clear and easy to follow, demonstrating a well-structured approach to enhancing translation performance in large language models.
The specialization dataset and evaluation framework will be useful for the community.

**Reasons To Reject:**

The approach of continued pretraining combined with instruction fine-tuning (as depicted in Figure 1) is quite standard in current LLM development. The paper lacks clarity on the specific technical contributions it offers beyond established practices.

---

> ### Author Rebuttal · Authors · 2024-05-30
>
> Thank you for your review and comments. We are glad that you found the methodology clear and easy to follow, and our dataset and evaluation framework useful for the community.
>
> > The approach of continued pretraining combined with instruction fine-tuning (as depicted in Figure 1) is quite standard in current LLM development. The paper lacks clarity on the specific technical contributions it offers beyond established practices.
>
> Regarding continued pretraining, while we recognize it is an established practice for adapting models to specific domains, its successful usage for expanding capabilities for several languages is less widespread.
>
> The most similar work to ours is ALMA, which proposes to CPT on monolingual data only and discards parallel data during CPT, advocating that large amounts of parallel data are no longer necessary for training translation systems and deeming this a “paradigm shift in machine translation”. We detach from them in the following ways:
>
> 1. Instead of disregarding parallel data, we include it as an extra training signal alongside monolingual data during CPT. We will make sure to emphasize this difference in the final version.
>
> 2. Through ablations on the recipe, we find that, contrary to the position put forward in ALMA, parallel data does help when mixed with monolingual data (see Figure 7).
>
> Furthermore, our technical developments go beyond continued pretraining by:
>
> 1. Curating a high-quality and diverse instruction-following dataset for translation-related tasks (going beyond translation only), which improves the performance of the trained model across the board;
>
> 2. Providing not only extensive experimentation and quantitative results, but also an analysis of relevant trends (e.g., impact of length of the source sentences on translation quality, proneness to edition in automatic post-editing);
>
> 3. Releasing all trained models, our instruction following dataset, our evaluation framework and test sets, and a collection of model generations to facilitate reproducibility and future analysis.
>
> We will make our contributions and technical developments clearer in the final version.

---

### Decision · Program_Chairs · 2024-07-10

**Decision:**

Accept

**Comment:**

The paper explores continued training of LLMs specifically for translation tasks. All reviewers agree to accept the submission. The method is straightforward, and the findings are helpful to the community. Although the technical contribution is limited, it is still a good study for MT.